# SCORE: A Unified Framework for Overshoot Refund in Online FDR Control

Qi Kuang [1]   Bowen Gang [1]   Yin Xia [1]

## Abstract

We propose a unified framework to enhance the power of online multiple hypothesis testing procedures based on $e$-values. While $e$-value-based methods offer robust online False Discovery Rate (FDR) control under minimal assumptions, they often suffer from power loss by discarding evidence that exceeds the rejection threshold. We address this inefficiency via the **S**equential **C**ontrol with **O**vershoot **R**efund for **E**-values (SCORE) framework, which leverages the inequality $\mathbb{I}(y \geq 1) \leq y - (y-1)_+$, valid for all $y \geq 0$, to reclaim this otherwise "wasted" evidence. This simple yet powerful insight yields a unified principle for improving a broad class of online testing algorithms. Building on this framework, we develop SCORE-enhanced versions of several state-of-the-art procedures, including SCORE-LOND, SCORE-LORD, and SCORE-SAFFRON, all of which strictly dominate their original counterparts while preserving valid finite-sample FDR control. Furthermore, under mild assumptions, SCORE permits retroactive updates of alpha-wealth by using the latest decision twice: first to determine its reward or loss, and then to refresh past wealth. Such a mechanism enables more aggressive testing strategies while maintaining valid FDR control, thereby further improving statistical power. The effectiveness of the proposed methods is validated through extensive simulation and real-data experiments.

## 1. Introduction

In the era of big data, scientific research and industrial decision-making are increasingly driven by large-scale, continuous experimentation. From pharmaceutical companies screening thousands of drug candidates (Blay et al., 2020)

to tech giants conducting ongoing A/B testing on user interfaces (Berman & Van den Bulte, 2022), hypotheses are frequently tested sequentially over time rather than in a single static batch. In this online setting, decisions for each hypothesis must be made immediately upon data acquisition, without knowledge of future tests or even the total number of hypotheses, and once made, these decisions cannot be revised. Consequently, the fundamental statistical challenge is to maximize the number of true discoveries while rigorously controlling the False Discovery Rate (FDR) (Benjamini & Hochberg, 1995) at every step of the testing process.

To address this challenge, the theoretical framework for online FDR control was pioneered by Foster & Stine (2008), who introduced the concept of *alpha-investing*. This approach treats the error budget $\alpha$ as a limited resource, or "wealth", that is allocated to individual tests and credited back upon rejections. This paradigm was significantly expanded by Aharoni & Rosset (2014) through the development of Generalized Alpha Investing (GAI). Building on this foundation, later work refined and strengthened the framework: Ramdas et al. (2017) proposed GAI++ to uniformly improve GAI; Javanmard & Montanari (2015; 2018) developed LOND and LORD to explicitly manage wealth recovery; and adaptive procedures such as SAFFRON (Ramdas et al., 2018) and ADDIS (Tian & Ramdas, 2019) were designed to enhance power by estimating the proportion of null hypotheses or accommodating conservative $p$-values. However, the validity of these widely used $p$-value-based methods typically relies on specific dependence assumptions, such as independence or Positive Regression Dependence on a Subset (PRDS) (Benjamini & Yekutieli, 2001). In complex, real-world data streams involving time-series or shared control groups, these assumptions are often violated, potentially compromising FDR control. More recently, Xu & Ramdas (2022) proposed SupLORD, which requires only conditional superuniformity. However, its primary focus is controlling the False Discovery Exceedance (FDX) rather than the FDR.

To circumvent the reliance on dependence assumptions, recent methodological advances have pivoted toward *e-values* as a robust alternative (Wang & Ramdas, 2022; Grünwald et al., 2024). Leveraging the property that an $e$-value's expectation is bounded by one under the null, Xu & Ramdas (2024) and Zhang et al. (2025) developed algorithms such

[1] Department of Statistics and Data Science, Fudan University, Shanghai, China. Correspondence to: Yin Xia <xiayin@fudan.edu.cn>.

*Proceedings of the 43rd International Conference on Machine Learning*, Seoul, South Korea. PMLR 306, 2026. Copyright 2026 by the author(s).

as e-LOND, e-LORD, and e-SAFFRON that maintain valid FDR control under minimal assumptions. Despite this attractive safety guarantee, these methods frequently suffer from reduced statistical power compared to their $p$-value counterparts. A key source of this inefficiency is the "overshoot" phenomenon: once an $e$-value crosses the rejection threshold, existing procedures make only a binary decision and ignore the remaining evidence carried by the magnitude of the $e$-value.

To improve statistical power, we propose a unified framework centered on the *Overshoot Refund* principle. The core intuition is straightforward: rather than treating the rejection threshold as a hard cutoff that discards excess evidence, we rigorously quantify this "overshoot" and refund it back to the algorithm's wealth budget. Our main contributions are as follows:

- We introduce the **S**equential **C**ontrol with **O**vershoot **R**efund for **E**-values (**SCORE**) framework. By exploiting a tighter bound on the false discovery proportion (FDP), we develop SCORE versions of state-of-the-art algorithms—specifically SCORE-LOND, SCORE-LORD, and SCORE-SAFFRON. We prove that these algorithms strictly dominate their original counterparts, offering uniformly higher power while maintaining valid finite-sample FDR control.

- We extend the theoretical analysis to enable **retroactive wealth updates**. Under mild dependence assumptions, the algorithm can leverage the latest decision to retroactively refresh the effective cost of past tests. This mechanism supports more aggressive procedures, denoted as SCORE$^+$, that further narrow the efficiency gap relative to traditional $p$-value-based methods.

**Conflict of Interest Disclosure**

The authors declare that they have no financial or non-financial conflicts of interest regarding the publication of this paper.

## 2. Problem Formulation

The online multiple testing problem can be formally described as follows. We consider an unlimited stream of null hypotheses $H_1, H_2, \ldots$ arriving sequentially over time. At each step $t \in \mathbb{N}$, the analyst must make a decision regarding the null hypothesis $H_t$ immediately, utilizing only the information available up to that point and without knowledge of future hypotheses or the total number of tests.

Let $\delta_t \in \{0, 1\}$ denote the decision for the $t$-th hypothesis, where $\delta_t = 1$ indicates that $H_t$ is rejected (a discovery), and $\delta_t = 0$ indicates a failure to reject. Let $\mathcal{H}_0 \subset \mathbb{N}$ denote the set of indices corresponding to true null hypotheses

and $\mathcal{H}_0(t) = \{1, \ldots, t\} \cap \mathcal{H}_0$. The cumulative number of rejections up to time $t$ is denoted by $R_t = \sum_{j=1}^{t} \delta_j$.

The primary objective is to control the error rate at any time $t$. The FDR is defined as the expected FDP where the FDP is the ratio of false rejections to total rejections:

$$\text{FDP}(t) = \frac{\sum_{j \in \mathcal{H}_0(t)} \delta_j}{R_t \vee 1}, \quad \text{and} \quad \text{FDR}(t) := \mathbb{E}[\text{FDP}(t)].$$

Given a pre-specified error budget $\alpha \in (0, 1)$, our goal is to ensure $\text{FDR}(t) \leq \alpha$ for all $t \geq 1$. Subject to this constraint, the secondary objective is to maximize average power (AP), i.e., $\text{AP}(t) = \mathbb{E}\left(\sum_{j \in \mathcal{H}_1(t)} \delta_j / |\mathcal{H}_1(t)|\right)$, where $\mathcal{H}_1(t) = \{1, \ldots, t\} \backslash \mathcal{H}_0(t)$.

To achieve robust control without relying on restrictive dependence assumptions between hypotheses, we focus on a testing framework based on $e$-*values*. Let $\mathcal{F}_{t-1}$ denote the $\sigma$-algebra generated by the history of decisions and observations prior to step $t$. An $e$-value $e_t$ for hypothesis $H_t$ is a non-negative random variable satisfying the following condition:

$$\text{if } H_t \text{ is true, then } \mathbb{E}[e_t \mid \mathcal{F}_{t-1}] \leq 1. \quad (1)$$

Under the alternative hypothesis, $e_t$ may take large values, reflecting evidence against the null. The online testing procedure using $e$-values proceeds as follows:

1. **Threshold Selection:** Before observing $e_t$, the algorithm selects a significance level (or rejection threshold) $\alpha_t \in (0, 1)$ based on the history $\mathcal{F}_{t-1}$. Formally, this means that the sequence $\{\alpha_t\}_{t \geq 1}$ is *predictable*, i.e., each $\alpha_t$ is measurable with respect to $\mathcal{F}_{t-1}$.

2. **Decision Rule:** Upon observing $e_t$, the null hypothesis $H_t$ is rejected if the evidence exceeds the reciprocal of the significance level:

$$\delta_t = \mathbb{I}(e_t \geq \alpha_t^{-1}).$$

This formulation is closely related to, but distinct from, the more familiar $p$-value-based online testing framework. In $p$-value procedures, small values provide evidence against the null and the rejection rule takes the form $p_t \leq \alpha_t$. In contrast, $e$-value procedures use large values as evidence and reject when $e_t \geq 1/\alpha_t$. The two notions are connected through calibrators. A $p$-value can be transformed into an $e$-value by a $p$-to-$e$ calibrator, while any $e$-value yields a valid $p$-value via $p_t = \min\{1, 1/e_t\}$ (Vovk & Wang, 2021). This conversion makes the rejection rules directly comparable: rejecting when $e_t \geq 1/\alpha_t$ is equivalent to rejecting when the calibrated $p$-value satisfies $p_t \leq \alpha_t$. This connection shows that $e$-value methods have a natural connection with classical $p$-value methods, rather than being a separate testing paradigm.

Under this decision rule, the trajectory of $R_t$ is fully determined by the fixed data stream and the sequence of thresholds $\{\alpha_j\}$ generated by that procedure. Accordingly, within the definition or analysis of any specific algorithm, $R_t$ refers to the rejection history generated by that algorithm.

## 3. An Improvement Framework via Overshoot

Many existing online testing algorithms can be viewed through a unified perspective: they construct a running estimator $\widehat{\text{FDP}}(t)$ for the FDP and dynamically adjust the significance level $\alpha_t$ to ensure this estimator remains bounded by $\alpha$. The FDP estimator in existing $e$-value-based online testing literature takes the following form:

$$\widehat{\text{FDP}}(t) = \sum_{j=1}^{t} \frac{C_j}{R_{j-1} + 1}. \tag{2}$$

We can interpret $C_j$ as the wealth invested for testing $H_j$. Different choices of $C_j$ lead to different procedures. For example, e-LOND and e-LORD both set $C_j = \alpha_j$, e-SAFFRON sets $C_j = \alpha_j \mathbb{I}\{e_j < 1/\lambda_j\}/(1-\lambda_j)$, where $\{\lambda_t\}_{t \geq 1} \in (0,1)$ is a predictable sequence of parameters.

A key tool used in proving the validity of these $e$-value-based procedures is the following indicator bound, which follows directly from Markov's inequality:

$$\delta_j = \mathbb{I}(e_j \geq \alpha_j^{-1}) \leq \alpha_j e_j. \tag{3}$$

However, the inequality (3) is loose whenever the evidence strictly exceeds the rejection threshold (i.e., $e_j > \alpha_j^{-1}$). In such cases, the "excess" evidence is effectively discarded, treating a marginal rejection the same as a highly confident one. This loss of information leads to conservative error estimates and, consequently, reduced statistical power.

Therefore, we propose to sharpen this bound by explicitly accounting for the excess evidence. We introduce the following elementary inequality, which forms the cornerstone of our framework.

**Lemma 3.1.** *For any $y \geq 0$,*

$$\mathbb{I}(y \geq 1) \leq y - (y-1)_+,$$

*where $(x)_+ = \max\{x, 0\}$.*

Applying Lemma 3.1 with $y = \alpha_j e_j$, we obtain a tighter bound on the decision indicator:

$$\delta_j \leq \alpha_j e_j - (\alpha_j e_j - 1)_+. \tag{4}$$

We define the second term above, referred to as the **overshoot** $O_j$, as the portion of the $e$-value that exceeds the rejection threshold:

$$O_j := (\alpha_j e_j - 1)_+.$$

Equation (4) implies that the actual cost in rejecting $H_j$ is not simply bounded by $\alpha_j e_j$, but can be reduced by the observed overshoot $O_j$. This allows us to "refund" this overshoot back to the error budget, thereby improving the power of the testing procedure without violating FDR control.

This observation motivates our proposed **S**equential **C**ontrol with **O**vershoot **R**efund for **E**-values (**SCORE**) framework. The intuition is straightforward: whenever an overshoot occurs, the standard cost assigned to a rejection exceeds the tighter bound on the false discovery indicator established in Lemma 3.1. We can therefore discount the cost charged to the algorithm by exactly $O_j$, effectively "refunding" this excess evidence. Specifically, we propose replacing the original invested wealth $C_j$ with the **SCORE-adjusted wealth**:

$$C_j^{\text{SCORE}} = (C_j - O_j)_+ = (C_j - (\alpha_j e_j - 1)_+)_+.$$

Here, the outer positive-part operator is applied to ensure the adjusted cost remains non-negative. Substituting this reduced cost into the FDP estimator lowers the estimated error, thereby creating more room in the error budget for future tests. This mechanism is formalized in the following general theorem, which provides a "plug-in" recipe to uniformly improve existing algorithms.

**Theorem 3.2.** *Suppose that $\forall j \in \mathcal{H}_0$, $\mathbb{E}[\alpha_j e_j \mid \mathcal{F}_{j-1}] \leq \mathbb{E}[C_j \mid \mathcal{F}_{j-1}]$. Then, $\forall j \in \mathcal{H}_0$, we have*

$$\mathbb{E}[\delta_j \mid \mathcal{F}_{j-1}] \leq \mathbb{E}[C_j^{\text{SCORE}} \mid \mathcal{F}_{j-1}].$$

*Consequently, the tighter estimator*

$$\widehat{\text{FDP}}_{\text{SCORE}}(t) = \sum_{j=1}^{t} \frac{C_j^{\text{SCORE}}}{R_{j-1} + 1}$$

*satisfies $\mathbb{E}[\widehat{\text{FDP}}_{\text{SCORE}}(t)] \geq \text{FDR}(t)$. Thus, any procedure ensuring $\widehat{\text{FDP}}_{\text{SCORE}}(t) \leq \alpha$ yields $\text{FDR}(t) \leq \alpha$.*

Since $C_j^{\text{SCORE}} \leq C_j$ almost surely, the SCORE estimator is uniformly no larger than the original estimator. Hence, any update rule derived from the SCORE framework is at least as powerful as its baseline counterpart, and is strictly more powerful whenever overshoots occur.

## 4. Applications: SCORE Algorithms

In this section, we demonstrate the practical power of the SCORE framework by applying it to three state-of-the-art $e$-value-based algorithms: e-LOND (Xu & Ramdas, 2024), e-LORD, and e-SAFFRON (Zhang et al., 2025). In the following, we refer to these SCORE variants as S-LOND, S-LORD, and S-SAFFRON when they appear in superscripts or subscripts. The definitions of $R_j$ and $O_j$ depend on the specific sequence $\alpha_j$ employed by each algorithm and therefore differ across methods. For brevity, we use a unified notation whenever no ambiguity arises.

## 4.1. SCORE-LOND

The e-LOND algorithm is the $e$-value analogue of the LOND procedure (Javanmard & Montanari, 2015). As mentioned in Section 3, e-LOND sets $C_j = \alpha_j$.

Applying the SCORE framework, we replace $C_j$ with

$$C_j^{\text{S-LOND}} = (\alpha_j - O_j)_+.$$

The resulting FDP estimator is therefore

$$\widehat{\text{FDP}}_{\text{S-LOND}}(t) = \sum_{j=1}^{t} \frac{C_j^{\text{S-LOND}}}{R_{j-1} + 1}.$$

Any predictable choice of significance levels $\{\alpha_j\}$ that keeps this estimator below $\alpha$ yields a valid procedure; the following proposition specifies one such choice.

**Proposition 4.1.** *Let $\{\gamma_t\}_{t \geq 1}$ be a predictable non-negative sequence such that $\sum_{t=1}^{\infty} \gamma_t = 1$. For $t \geq 2$, define*

$$\alpha_t^{\text{S-LOND}} = \gamma_t(R_{t-1} + 1)\left(\alpha + \sum_{j=1}^{t-1} \frac{\min(O_j, \alpha_j^{\text{S-LOND}})}{R_{j-1} + 1}\right), \tag{5}$$

*and set $\alpha_1^{\text{S-LOND}} = \gamma_1 \alpha$. This updating rule ensures that $\widehat{\text{FDP}}_{\text{S-LOND}}(t) \leq \alpha$ for all $t$. Consequently, the decision rule $\delta_t = \mathbb{I}(e_t \geq 1/\alpha_t^{\text{S-LOND}})$ satisfies $\text{FDR}(t) \leq \alpha$ for all $t$.*

To appreciate the improvement, recall that the original e-LOND algorithm (Xu & Ramdas, 2024) defines its significance levels as:

$$\alpha_t^{\text{e-LOND}} = \alpha \gamma_t(R_{t-1}^{\text{e-LOND}} + 1).$$

Comparing this with the SCORE-LOND update rule in (5), we can interpret the term $\min(O_j, \alpha_j^{\text{S-LOND}})$ as a "refund" recovered from the $j$-th test. Unlike the standard method, which discards this excess evidence, SCORE accumulates it to boost future testing power. Since these refunds are non-negative, the SCORE-LOND threshold is uniformly superior, as established by the following proposition.

**Proposition 4.2.** *For any $t \geq 1$, $\alpha_t^{\text{S-LOND}} \geq \alpha_t^{\text{e-LOND}}$. Consequently, SCORE-LOND is a strict improvement over e-LOND if $e_j > 1/\alpha_j^{\text{S-LOND}}$ for some $j < t$.*

## 4.2. SCORE-LORD

The e-LORD algorithm (Zhang et al., 2025) is the $e$-value analogue of the LORD procedure (Javanmard & Montanari, 2015; 2018), and it generalizes e-LOND by allowing for a dynamic, history-dependent allocation of the error budget. Despite this added flexibility, e-LORD utilizes the same base cost $C_j = \alpha_j$. Consequently, the application of the SCORE framework mirrors the e-LOND case exactly: we substitute the fixed cost with the adjusted cost $C_j^{\text{S-LORD}} = (\alpha_j - O_j)_+$.

The distinction lies solely in the update rule. While e-LOND uses a fixed sequence $\gamma_t$, e-LORD employs a predictable weighting sequence $\omega_t$ to manage the available $\alpha$-wealth. More specifically, e-LORD uses the following update rule:

$$\alpha_t^{\text{e-LORD}} = \omega_t(R_{t-1}^{\text{e-LORD}} + 1)\left(\alpha - \sum_{j=1}^{t-1} \frac{\alpha_j^{\text{e-LORD}}}{R_{j-1}^{\text{e-LORD}} + 1}\right),$$

where $\omega_t \in (0, 1)$ is a predictable sequence.

The following proposition establishes the proposed SCORE-type update rule and the dominance of SCORE-LORD.

**Proposition 4.3.** *Given a predictable sequence of weighting parameters $\{\omega_t\}_{t \geq 1} \in (0, 1)$, define the update rule for $t \geq 2$:*

$$\alpha_t^{\text{S-LORD}} = \omega_t(R_{t-1} + 1)\left(\alpha - \sum_{j=1}^{t-1} \frac{(\alpha_j^{\text{S-LORD}} - O_j)_+}{R_{j-1} + 1}\right),$$

*and set $\alpha_1^{\text{S-LORD}} = \omega_1 \alpha$. Then the decision rule $\delta_t = \mathbb{I}(e_t \geq 1/\alpha_t^{\text{S-LORD}})$ satisfies $\text{FDR}(t) \leq \alpha$ for all $t$. Furthermore, for all $t \geq 1$, $\alpha_t^{\text{S-LORD}} \geq \alpha_t^{\text{e-LORD}}$, therefore SCORE-LORD is a strict improvement over e-LORD if $e_j > 1/\alpha_j^{\text{S-LORD}}$ for some $j < t$.*

## 4.3. SCORE-SAFFRON

The e-SAFFRON algorithm (Zhang et al., 2025) is the $e$-value analogue of the SAFFRON procedure (Ramdas et al., 2018). It improves upon LORD by incorporating the concept of "candidate" hypotheses. The algorithm utilizes a predictable sequence of parameters $\lambda_j \in (0, 1)$, which serves as a screening threshold. A hypothesis is considered a *candidate* for rejection if its evidence is sufficiently large ($e_j \geq \lambda_j^{-1}$). The core insight, tracing back to the Storey-BH procedure (Storey, 2002), is that one should only pay an error cost for hypotheses that fail to become candidates.

Recall that the e-SAFFRON uses following cost in (2):

$$C_j = \frac{\alpha_j \mathbb{I}\{e_j < \lambda_j^{-1}\}}{1 - \lambda_j}.$$

Here, the algorithm pays a penalty proportional to $\alpha_j$ only when the evidence is weak ($e_j < \lambda_j^{-1}$). If the evidence is strong enough to pass the candidate threshold, the cost is zero.

The SCORE framework allows us to improve this cost function in two steps:

1. **Tightening the Candidate Penalty:** We replace the binary indicator $\mathbb{I}\{e_j < \lambda_j^{-1}\} = 1 - \mathbb{I}\{\lambda_j e_j \geq 1\}$ with the tighter continuous bound derived from Lemma 3.1. This yields an intermediate cost $C_j' = \alpha_j(1 - \lambda_j e_j + (\lambda_j e_j - 1)_+)/(1 - \lambda_j) \leq C_j$, which strictly improves e-SAFFRON even without considering overshoots.

2. **Refunding the Overshoot:** We then apply the SCORE principle to this tighter baseline by subtracting the rejection overshoot $O_j$.

Combining these steps yields the SCORE-SAFFRON adjusted cost:

$$C_j^{\text{S-SAF}} = \left( \frac{\alpha_j(1 - \lambda_j e_j + (\lambda_j e_j - 1)_+)}{1 - \lambda_j} - O_j \right)_+$$
$$= \left( \frac{\alpha_j(1 - \lambda_j e_j)}{1 - \lambda_j} - O_j \right)_+.$$

Using this cost, we construct the FDP estimator

$$\widehat{\text{FDP}}_{\text{S-SAF}}(t) = \sum_{j=1}^{t} \frac{C_j^{\text{S-SAF}}}{R_{j-1} + 1}.$$

The following proposition establishes the update rule for $\alpha_t$ that ensures $\widehat{\text{FDP}}_{\text{S-SAF}}(t) \leq \alpha$ and its dominance over the standard e-SAFFRON updating rule.

**Proposition 4.4.** *Given predictable sequences of weights $\{\omega_t\}_{t \geq 1}$ and candidate thresholds $\{\lambda_t\}_{t \geq 1}$ in $(0, 1)$, define the update rule for $t \geq 2$:*

$$\alpha_t^{\text{S-SAF}} = \omega_t(1 - \lambda_t)(R_{t-1} + 1) \left( \alpha - \sum_{j=1}^{t-1} \frac{C_j^{\text{S-SAF}}}{R_{j-1} + 1} \right),$$

*and set $\alpha_1^{\text{S-SAF}} = \omega_1(1 - \lambda_1)\alpha$. Then the decision rule $\delta_t = \mathbb{I}(e_t \geq 1/\alpha_t^{\text{S-SAF}})$ satisfies $\text{FDR}(t) \leq \alpha$ for all $t$. Furthermore, for all $t \geq 1$, $\alpha_t^{\text{S-SAF}} \geq \alpha_t^{\text{SAF}}$, where*

$$\alpha_t^{\text{SAF}} = \omega_t(1 - \lambda_t)(R_{t-1}^{\text{SAF}} + 1)$$
$$\times \left( \alpha - \sum_{j=1}^{t-1} \frac{\alpha_j^{\text{SAF}}\mathbb{I}\{e_j < 1/\lambda_j\}}{(1 - \lambda_j)(R_{j-1}^{\text{SAF}} + 1)} \right)$$

*is the standard updating rule for e-SAFFRON. Consequently, SCORE-SAFFRON is a strict improvement over e-SAFFRON if $1/\alpha_j^{\text{SAF}} \geq e_j > 1/\alpha_j^{\text{S-SAF}}$ or $\lambda_j^{-1} > e_j > 0$ for some $j < t$.*

This result highlights the dual benefit of SCORE-SAFFRON: it saves budget on non-candidates by measuring *how far* they are from the threshold, and it also recovers budget from successful rejections via the refund mechanism.

More broadly, the gains from SCORE depend on the quality of the available $e$-values. If all alternative $e$-values are extremely large, the original procedure already rejects most alternatives and the room for improvement is limited. If all $e$-values are uniformly conservative and never cross the rejection threshold, then little overshoot can be refunded and SCORE essentially reduces to the baseline method. The most favorable regime is the common mixed-signal

setting: some alternatives generate strong evidence and hence release refunds, while other alternatives are close to the rejection threshold and can be detected only after additional wealth is made available.

### 4.4. Connection to Existing Work

Our proposed SCORE framework addresses a fundamental inefficiency in online hypothesis testing: the discrepancy between the actual error committed and the theoretical cost charged by the algorithm. In the domain of $p$-value-based methods, this issue is often tackled by exploiting the super-uniformity of null $p$-values, as seen in ADDIS (Tian & Ramdas, 2019) and the Super-Uniformity Reward (SUR) framework (Döhler et al., 2024). While sharing our motivation of wealth recovery, these approaches generally necessitate specialized algorithmic designs or specific distributional assumptions. In contrast, SCORE utilizes a simple elementary inequality to serve as a universal "plug-in" module, uniformly improving existing $e$-value algorithms—such as e-LOND, e-LORD, and e-SAFFRON—without requiring fundamental structural changes to the testing procedure.

Conceptually, our work finds its most direct antecedents in the structure-adaptive algorithms SAST (Gang et al., 2023) and SMART (Wang et al., 2024). These methods address the "overshoot" problem by utilizing the conditional local false discovery rate (CLfdr) to quantify the precise cost of a rejection, effectively recycling the "gain" from strong signals back into the budget. However, their theoretical validity relies on the consistent estimation of the CLfdr, a computationally intensive task that typically yields only asymptotic error control. The SCORE framework bridges this gap: it captures the power benefits of the overshoot refund mechanism pioneered by SAST and SMART, but by operating within the $e$-value paradigm, it bypasses the need for density estimation and guarantees rigorous finite-sample FDR control.

The $e$-value-based online FDR literature is still relatively young. To the best of our knowledge, e-LOND, e-LORD, and e-SAFFRON are the canonical finite-sample FDR procedures currently available for this setting, and they mirror the central LOND/LORD/SAFFRON families in the $p$-value literature. Our contribution is therefore not a new isolated rule for one algorithm, but a systematic refinement that can be applied whenever an online $e$-value method proves validity through this standard Markov-derived indicator bound.

## 5. Retroactive Wealth Updates

In the preceding sections, we addressed one source of power loss—the "overshoot"—while relying solely on the conditional $e$-validity assumption (1). Under this minimal assumption, validity necessitates the use of the denominator

$R_{j-1} + 1$ in the FDP estimator (2). As discussed by Xu & Ramdas (2024) and Zhang et al. (2025), this choice acts as a predictable lower bound for the true number of rejections $R_t$. Since $R_{j-1} + 1 \leq R_t \vee 1$ for any rejected hypothesis $j \leq t$, this substitution ensures that the estimator never underestimates the error, thereby guaranteeing validity without requiring any further assumptions on the dependence structure.

However, this robustness comes with a significant efficiency cost. In the $p$-value-based online testing literature (Javanmard & Montanari, 2018; Ramdas et al., 2017; 2018; Tian & Ramdas, 2019; Zrnic et al., 2021), the FDP is typically estimated using a global denominator of the form:

$$\widehat{\text{FDP}}(t) = \frac{\sum_{j=1}^{t} C_j}{R_t \vee 1}. \tag{6}$$

The estimator in (6) is intuitively tighter because it scales the accumulated cost by the *current* total number of discoveries, rather than by the number of discoveries at the time each decision was made.

The distinction between these two denominators implies a fundamental difference in how the "cost" of a discovery is accounted for.

- **Fixed Cost** ($R_{j-1} + 1$): When using the local denominator, the contribution of the $j$-th hypothesis to the FDP estimate is fixed at $\frac{C_j}{R_{j-1}+1}$ at the moment the decision is made and remains unchanged, regardless of any subsequent discoveries.

- **Retroactive Update** ($R_t \vee 1$): When using the global denominator, the contribution of the $j$-th hypothesis is $\frac{C_j}{R_t \vee 1}$. As the algorithm proceeds and additional discoveries are made (so that $R_t$ increases), the effective cost assigned to *past* decisions is retroactively reduced.

We term this latter mechanism a **retroactive wealth update**. Effectively, a new discovery at time $t$ not only earns an immediate reward but also retroactively reduces the penalty of all prior discoveries, freeing up alpha-wealth that was previously locked to cover conservative error estimates.

A natural question arises: under what conditions can we safely employ this aggressive retroactive strategy within the $e$-value framework? Although the local denominator $R_{j-1} + 1$ is strictly necessary when relying solely on conditional $e$-validity, we demonstrate that a mild and intuitive assumption is sufficient to validate the global denominator $R_t \vee 1$. This adjustment allows us to further improve power.

### 5.1. Conditional Positive Quadrant Dependence

We first introduce the key assumption that enables the validity of our proposed retroactive updates.

**Assumption 5.1.** Fix a terminal time $t$. For any null index $j \in \mathcal{H}_0(t)$, conditioned on the history $\mathcal{F}_{j-1}$, the pair $(e_j, R_t)$ is Positive Quadrant Dependent (PQD, Lehmann (2011)). Here $e_j$ is the current evidence, whereas $R_t$ is the total rejections at time $t$. That is, for any $x, y$:

$$\mathbb{P}(e_j \leq x, R_t \leq y \mid \mathcal{F}_{j-1}) \geq \mathbb{P}(e_j \leq x \mid \mathcal{F}_{j-1}) \\ \times \mathbb{P}(R_t \leq y \mid \mathcal{F}_{j-1}),$$

which is equivalent to saying that for any non-decreasing functions $f, g : \mathbb{R} \to \mathbb{R}$, we have

$$\text{Cov}(f(e_j), g(R_t) \mid \mathcal{F}_{j-1}) \geq 0.$$

We refer to Assumption 5.1 as the Conditional Positive Quadrant Dependence (CPQD) assumption. Intuitively, CPQD says that a larger current $e$-value should not make the final number of discoveries smaller. This aligns with the alpha-wealth mechanism: a large observed $e$-value $e_j$ is more likely to trigger a rejection at step $j$, directly increasing $R_t$; even when its main effect is through an overshoot refund, it preserves more alpha-wealth for future tests. The resulting larger wealth leads to more permissive future thresholds, making additional discoveries more likely. Thus, in the regimes targeted by SCORE$^+$, $e_j$ and $R_t$ tend to move in the same direction.

Importantly, CPQD is only needed to justify the retroactive denominator used by SCORE$^+$; the basic SCORE algorithms in Section 4 remain valid under conditional $e$-validity alone. Appendix A shows that CPQD follows from independent $e$-values together with monotone threshold updates, a standard sufficient condition in online testing. This sufficient condition covers the canonical monotone alpha-investing setting and supports viewing CPQD as a mild dependence requirement for the retroactive update.

While the CPQD assumption enables upgrades to all of the algorithms discussed above, the remainder of this section focuses on the LORD and SAFFRON variants, denoted as SCORE$^+$. We omit the explicit derivation of SCORE$^+$-LOND, given that e-LOND is a special case of e-LORD (Zhang et al., 2025), and SCORE$^+$-LOND therefore follows directly as the corresponding special case of the SCORE$^+$-LORD procedure developed below.

### 5.2. SCORE$^+$-LORD

We define the **SCORE$^+$-LORD** procedure by substituting the local denominator $R_{j-1} + 1$ with the global denominator $R_t \vee 1$. The corresponding FDP estimator is given by:

$$\widehat{\text{FDP}}_{\text{S}^+\text{-LORD}}(t) = \frac{1}{R_t \vee 1} \sum_{j=1}^{t} (\alpha_j - O_j)_+.$$

The following theorem justifies the use of this tighter estimator for online FDR control.

**Theorem 5.2.** *Under Assumption 5.1, we have:*

$$\text{FDR}(t) \leq \mathbb{E}\left[\widehat{\text{FDP}}_{\text{S}^+\text{-LORD}}(t)\right].$$

*Consequently, any procedure that ensures $\widehat{\text{FDP}}_{\text{S}^+\text{-LORD}}(t) \leq \alpha$ almost surely for all $t$ guarantees online FDR control at level $\alpha$.*

Having established the validity conditions, we now derive the corresponding update rule. The key feature of SCORE$^+$-LORD is its dynamic wealth updating mechanism: as the number of rejections increases, the effective cost of past tests decreases, thereby releasing additional budget for future tests.

**Proposition 5.3.** *Given a predictable sequence of weighting parameters $\{\omega_t\}_{t\geq 1}$ where $\omega_t \in (0,1)$, the SCORE$^+$-LORD significance level is set as $\alpha_1^{\text{S}^+\text{-LORD}} = \omega_1\alpha$ and*

$$\alpha_t^{\text{S}^+\text{-LORD}} = \omega_t(R_{t-1} \vee 1)W_t, \quad t \geq 2, \qquad (7)$$

*where the remaining wealth $W_t$ is calculated as:*

$$W_t = \alpha - \sum_{j=1}^{t-1} \frac{(\alpha_j^{\text{S}^+\text{-LORD}} - O_j)_+}{R_{t-1} \vee 1}.$$

*This update rule ensures that $\widehat{\text{FDP}}_{\text{S}^+\text{-LORD}}(t) \leq \alpha$ for all $t$.*

To see the power of this approach, we compare the wealth $W_t$ in (7) with that of the standard SCORE-LORD algorithm (Proposition 4.3). In the SCORE version, the cost of the $j$-th test is fixed at $(\alpha_j^{\text{S-LORD}} - O_j)_+/(R_{j-1} + 1)$ at the time it is incurred. In contrast, in SCORE$^+$-LORD, the cost is re-evaluated at every step $t$ as $(\alpha_j^{\text{S-LORD}} - O_j)_+/(R_{t-1} \vee 1)$. As the algorithm accumulates more discoveries, $R_{t-1}$ increases, leading to a systematic reduction in the effective cost assigned to all tests with $j < t$. This retroactive cost reduction releases previously locked alpha-wealth back into the budget $W_t$, thereby enabling more aggressive testing in subsequent steps.

### 5.3. SCORE$^+$-SAFFRON

We now extend this logic to the SAFFRON framework, combining the benefits of candidate screening, overshoot refund, and retroactive updates.

Formally, given a predictable sequence of candidate thresholds $\{\lambda_j\}_{j\geq 1} \in (0,1)$, define

$$\widehat{\text{FDP}}_{\text{S}^+\text{-SAF}}(t) = \frac{1}{R_t \vee 1} \sum_{j=1}^{t} C_j^{\text{S}^+\text{-SAF}},$$

where

$$C_j^{\text{S}^+\text{-SAF}} = \left(\frac{\alpha_j(1 - \lambda_j e_j)}{1 - \lambda_j} - O_j\right)_+.$$

The following Theorem gives an update rule to ensure $\widehat{\text{FDP}}_{\text{S}^+\text{-SAF}}(t)$ is bounded by $\alpha$.

**Theorem 5.4.** *The SCORE$^+$-SAFFRON update rule $\alpha_1^{\text{S}^+\text{-SAF}} = \omega_1(1 - \lambda_1)\alpha$ and*

$$\alpha_t^{\text{S}^+\text{-SAF}} = \omega_t(1 - \lambda_t)(R_{t-1} \vee 1)W_t, \quad t \geq 2, \qquad (8)$$

$$W_t = \alpha - \sum_{j=1}^{t-1} \frac{C_j^{\text{S}^+\text{-SAF}}}{R_{t-1} \vee 1},$$

*ensures $\widehat{\text{FDP}}_{\text{S}^+\text{-SAF}}(t) \leq \alpha$ for all $t$. Furthermore, under Assumption 5.1, we have*

$$\text{FDR}(t) \leq \mathbb{E}\left[\widehat{\text{FDP}}_{\text{S}^+\text{-SAF}}(t)\right].$$

*Consequently, the decision rule $\delta_t = \mathbb{I}(e_t \geq 1/\alpha_t^{\text{S}^+\text{-SAF}})$ satisfies $\text{FDR}(t) \leq \alpha$ for all $t$.*

## 6. Numerical Experiments

In this section we evaluate the empirical performance of the proposed SCORE and SCORE$^+$ procedures against their standard counterparts. For each synthetic experiment, we compute empirical FDR and average power over 500 repetitions at each time point. The main figures focus on average power and power ratios, while the corresponding FDR trajectories are reported in Appendix B.1. The power ratio is computed pointwise in time relative to the corresponding baseline within the same algorithmic family: e-LORD for LORD-type procedures and e-SAFFRON for SAFFRON-type procedures. Thus, a ratio above one directly indicates the additional power gained from the SCORE accounting rule. More comparisons to $p$-value-based methods are deferred to Section E, and sensitivity to the target FDR level is summarized in Appendix B.3.

### 6.1. Independent Setting: Gaussian Mixture Model

We begin by considering an independent setting where the data stream is generated from a Gaussian mixture model. We sequentially observe $X_1, \ldots, X_T$ with a time horizon of $T = 1000$. The underlying states and observations are generated as follows:

$$\theta_t \overset{i.i.d.}{\sim} \text{Ber}(\pi_1), \quad \mu_t \mid \theta_t \sim (1 - \theta_t)\delta_0 + \theta_t N(3,5),$$

$$X_t \mid \mu_t \sim N(\mu_t, 1),$$

where $\delta_0$ is the point mass at 0. The goal is to test the sequence of null hypotheses $H_t : \mu_t = 0$. In this experiment, we consider an idealized scenario where both the null and alternative densities are known. Consequently, we employ the likelihood ratio as the $e$-value:

$$e_t = \frac{\phi(X_t; 3, 6)}{\phi(X_t; 0, 1)},$$

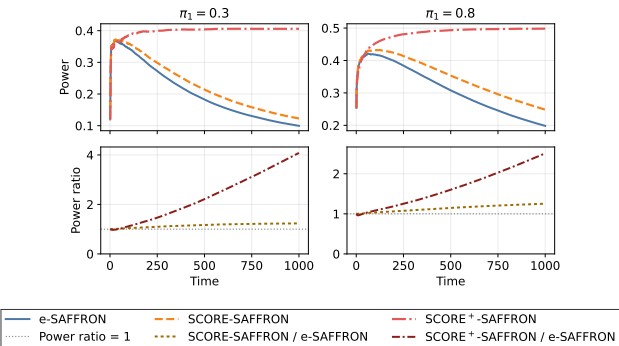

*Figure 1.* Independent setting, SAFFRON family. Power and power ratios relative to e-SAFFRON are shown for $\pi_1 \in \{0.3, 0.8\}$.

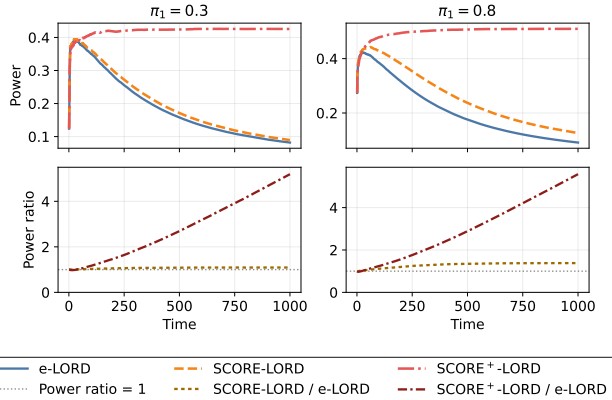

*Figure 2.* Independent setting, LORD family. Power and power ratios relative to e-LORD are shown for $\pi_1 \in \{0.3, 0.8\}$.

where $\phi(\cdot; \mu, \sigma^2)$ denotes the density function of a normal distribution with mean $\mu$ and variance $\sigma^2$. The target FDR level is set at $\alpha = 0.05$. For LORD-type methods we set $\omega_t \equiv 0.05$. For SAFFRON-type methods, we additionally fix $\lambda_t \equiv 0.5$ as suggested by Ramdas et al. (2018). We set $\pi_1 \in \{0.3, 0.8\}$. The average power results are shown in Figures 1 and 2.

As shown by the FDR trajectories in Appendix B.1, all six methods control the online FDR for both values of $\pi_1$. We therefore focus mainly on the power comparison. Across all settings, the SCORE$^+$ variants achieve the highest power, followed by the standard SCORE procedures and then the original baselines. The power-ratio panels show that standard SCORE consistently improves over the baseline, with moderate but persistent gains from refunding excess evidence.

The gains are substantially larger for SCORE$^+$. By retroactively updating past costs using the current number of discoveries, SCORE$^+$ releases additional wealth as discoveries accumulate, so its relative advantage grows over time.

## 6.2. Dependent Setting: Autoregressive Exponential Model

Next, we evaluate the procedures in a dependent setting. We simulate a stream of $T = 1000$ observations $X_1, \ldots, X_T$, where the conditional distribution of $X_t$ depends on $X_{t-1}$. Specifically, we generate the data as follows:

$$\theta_t \overset{i.i.d.}{\sim} \text{Bernoulli}(\pi_1),$$
$$\eta_t = 1 + \rho X_{t-1} \quad (\text{with } X_0 = 0),$$
$$\mu_t \overset{i.i.d.}{\sim} 0.5\delta_3 + 0.5\delta_{20}$$
$$X_t \mid X_{t-1}, \theta_t, \mu_t \sim (1 - \theta_t)\text{Exp}(\eta_t) + \theta_t\text{Exp}\left(\frac{\eta_t}{\mu_t}\right).$$

Here, $\theta_t \in \{0, 1\}$ indicates the truth of the hypothesis ($\theta_t = 1$ if non-null, 0 otherwise). The baseline rate parameter $\eta_t$ depends on the previous observation, inducing temporal dependence with correlation coefficient $\rho = 0.5$. Under the alternative ($\theta_t = 1$), the signal strength $\mu_t$ is drawn uniformly from $\{3, 20\}$, scaling the mean of the distribution by a factor of $\mu_t$. We set the non-null probability $\pi_1 \in \{0.3, 0.8\}$. The null hypotheses are $\{H_t : \theta_t = 0\}$.

We utilize conditional likelihood-ratio $e$-values, which are constructed to be valid under the specific dependence structure of the data stream:

$$e_t = \frac{1}{3} \exp\left(\eta_t X_t \left(1 - \frac{1}{3}\right)\right).$$

This construction remains conditionally valid under the null because, given the past, the null distribution of $X_t$ is $\text{Exp}(\eta_t)$ and the displayed likelihood ratio has conditional mean one. The mixture over $\mu_t \in \{3, 20\}$ creates a mixed-signal regime: moderate alternatives are closer to the rejection boundary, whereas strong alternatives tend to generate large $e$-values and hence sizeable overshoots. This setting is therefore well suited for assessing whether refunded wealth from strong discoveries can improve the detection of later or weaker signals under temporal dependence.

The target FDR level is set at $\alpha = 0.05$. For both LORD type methods and SAFFRON type methods, we use the adaptive Rejection-Adjusted Investment (RAI) weighting from Zhang et al. (2025) with parameters $\omega_1 = 0.05$ and $\varphi = \psi = 0.5$. The explicit update rule is $\omega_{t+1} = \omega_1 + \omega_1 \left(\sum_{j=1}^{t-R_t} \varphi^j - \sum_{j=1}^{R_t} \psi^j\right)$. For SAFFRON type methods we additionally set $\lambda_t \equiv 0.5$. The average power results are presented in Figures 3 and 4.

The FDR trajectories in Appendix B.1 show that all methods maintain valid FDR control. The power and power-ratio panels show that the same qualitative ordering from the independent experiment persists under temporal dependence: SCORE improves over the corresponding baseline,

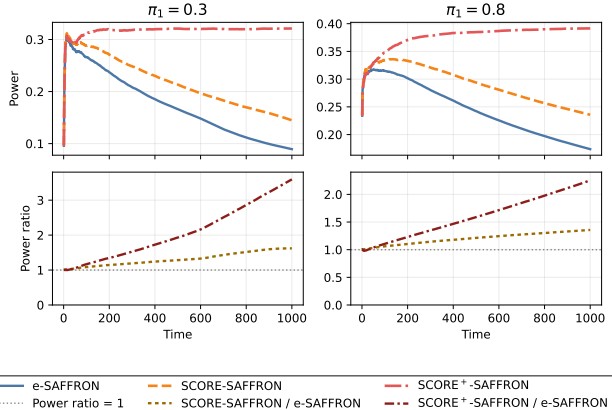

*Figure 3.* Dependent setting, SAFFRON family. Power and power ratios relative to e-SAFFRON are shown for $\pi_1 \in \{0.3, 0.8\}$.

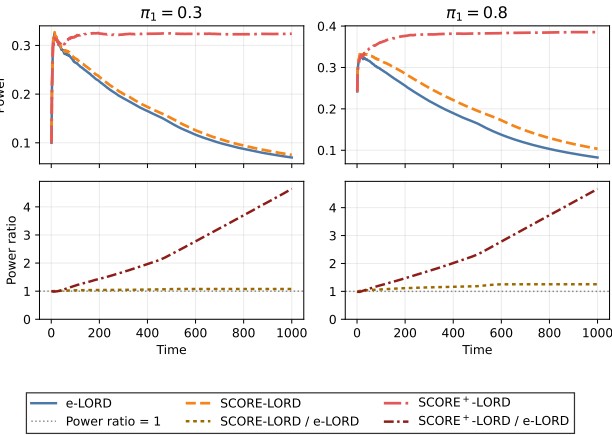

*Figure 4.* Dependent setting, LORD family. Power and power ratios relative to e-LORD are shown for $\pi_1 \in \{0.3, 0.8\}$.

and SCORE$^+$ yields the largest gain. The standard SCORE ratios stay above one for both the LORD and SAFFRON families, indicating that the overshoot refund remains beneficial even when the observations are dependent. The gain is particularly visible for SAFFRON, where the sharper SCORE inequality is used in two places: first to tighten the candidate penalty and then to refund the rejection overshoot.

### 6.3. Phenotypic Growth Prediction for Yeast

In this section, we demonstrate the practical utility of the SCORE framework using a real-world dataset from a high-throughput chemical genomics screen (Wildenhain et al., 2016). This study investigates 417,026 chemical-genetic interactions sequentially. For each chemical compound, growth defects across various yeast deletion strains are measured to identify significant strain-compound interactions indicative of genetic sensitivity.

Since the original data are reported as $p$-values, we convert them into valid $e$-values using the standard calibration $e =$

$\frac{1-p+p\log(p)}{p(-\log(p))^2}$, following Vovk & Wang (2021). We then apply both the proposed SCORE algorithms and standard baseline methods to this sequential data stream, targeting an FDR level of $\alpha = 0.1$.

Table 1 summarizes the number of discoveries achieved by each method. The results strongly corroborate the efficacy of the SCORE framework. Both SCORE-LORD (28,808 discoveries) and SCORE-SAFFRON (37,748 discoveries) detect substantially more interactions than their standard counterparts, e-LORD (25,954) and e-SAFFRON (29,867). Notably, the SCORE$^+$ variants achieve the largest gains: SCORE$^+$-LORD detects 44,210 interactions, nearly doubling the yield of e-LORD, while SCORE$^+$-SAFFRON identifies 43,624 significant interactions. These findings highlight that recovering "wasted" evidence via the combined mechanisms of overshoot refund and retroactive wealth updates is highly effective in large-scale, high-throughput screening scenarios.

*Table 1.* Number of discoveries made by various online FDR procedures for the Wildenhain et al. (2016) dataset; $\alpha = 0.1$.

| Type | Method | Discoveries |
|------|--------|-------------|
| LORD-type | e-LORD | 25954 |
| | SCORE-LORD | 28808 |
| | SCORE$^+$-LORD | 44210 |
| SAFFRON-type | e-SAFFRON | 29867 |
| | SCORE-SAFFRON | 37748 |
| | SCORE$^+$-SAFFRON | 43624 |

## 7. Conclusion

In this work, we introduced the SCORE framework for online multiple testing, built on the elementary yet powerful inequality $\mathbb{I}(y \geq 1) \leq y - (y - 1)_+$. By sharpening the standard indicator bound derived from Markov's inequality, we established a rigorous mechanism to recover the "overshoot"—the excess evidence from strong rejections that is typically discarded by existing methods. We demonstrated that converting this overshoot into a refundable error budget yields uniformly more powerful variants of state-of-the-art algorithms, specifically e-LOND, e-LORD, and e-SAFFRON, while maintaining valid FDR control.

The same plug-in idea is compatible with long-horizon online error criteria, such as decaying-memory FDR; we outline this extension in Appendix C. More broadly, because the crude inequality $\mathbb{I}(y \geq 1) \leq y$ is widely employed in validity proofs throughout the $e$-value literature, a wide range of $e$-value-based procedures, including those in offline settings, could potentially be enhanced by explicitly accounting for the overshoot.

## Acknowledgements

We sincerely thank the anonymous reviewers and the area chair for their insightful comments and constructive suggestions, which have greatly improved the quality of this manuscript. This work was supported by the Key Program of the National Natural Science Foundation of China (Grant No. 12331009).

## Impact Statement

This paper presents work whose goal is to advance the field of Machine Learning. There are many potential societal consequences of our work, none which we feel must be specifically highlighted here.

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

# A. Validity under Independence and Monotonicity

In this section, we show that both Assumption 5.1 (for $e$-values) and its counterpart for $p$-values are implied by familiar independence-plus-monotonicity conditions. This result builds a bridge between the existing literature and our abstract assumptions: while independence and monotonicity suffice to guarantee our conditions, the assumptions we deploy are strictly weaker and do not require full independence. In short, our theoretical conditions are compatible with standard sufficient conditions from prior work but remain valid under substantially milder dependence structures.

## A.1. Sufficiency for $e$-value-based Methods

**Proposition A.1.** *Suppose the $e$-values $\{e_j\}_{j=1}^{t}$ are mutually independent. If the significance levels $\alpha_k$ are coordinate-wise non-decreasing functions of the past decisions $(\delta_1, \ldots, \delta_{k-1})$, then Assumption 5.1 holds.*

We now verify the monotonicity condition for our proposed update rules, completing the validity argument for SCORE$^+$-LORD and SCORE$^+$-SAFFRON.

**Proposition A.2** (Monotonicity of SCORE$^+$ Update Rules). *Let $\omega_t \in (0,1)$ and $\lambda_t \in (0,1)$ be deterministic sequences that may depend on $t$ but not on the rejection history $(\delta_1, \ldots, \delta_{t-1})$. Then, the significance levels $\alpha_t$ defined by (7) and by (8) are coordinate-wise non-decreasing in the rejection history.*

**Corollary A.3.** *Suppose the $e$-values are mutually independent. Then, by Proposition A.1 and A.2, Assumption 5.1 holds, and hence the SCORE$^+$ procedures control FDR at level $\alpha$.*

*Remark* A.4. The significance level update rules need not strictly follow the forms of (7) or (8); any update rule that is coordinate-wise non-decreasing and ensures the FDP estimator is upper bounded by $\alpha$ suffices. For instance, the rule presented in Javanmard & Montanari (2018) and Ramdas et al. (2018) can be adopted.

## A.2. Extension to $p$-value-based Methods

The dependence perspective developed for $e$-values extends naturally to $p$-value-based methods. Just as CPQD serves as a sufficient condition for $e$-value algorithms, its counterpart, Conditional Negative Quadrant Dependence (CNQD), suffices for $p$-value algorithms. The key distinction lies in the direction of dependence: while $e$-values and future rejections are positively associated (both large values favor rejections), $p$-values and future rejections are negatively associated (small $p$-values lead to more discoveries). This duality is reflected in the switch from PQD to NQD.

For $p$-value-based methods, we also require an additional validity condition on the null hypotheses: each null $p$-value must be *conditionally super-uniform*, i.e., $\mathbb{P}(p_j \leq \alpha \mid \mathcal{F}_{j-1}) \leq \alpha$ for all $\alpha \in [0,1]$. This condition replaces the conditional $e$-validity requirement $\mathbb{E}[e_j \mid \mathcal{F}_{j-1}] \leq 1$ used for $e$-values.

We introduce the key assumption for $p$-value-based methods.

**Assumption A.5** (Conditional Negative Quadrant Dependence). For any null index $j \in \mathcal{H}_0$, conditioned on the history $\mathcal{F}_{j-1}$, the pair $(p_j, R_t)$ is Negatively Quadrant Dependent (NQD). That is, for any $x, y \in \mathbb{R}$:

$$\mathbb{P}(p_j \leq x, R_t \leq y \mid \mathcal{F}_{j-1}) \leq \mathbb{P}(p_j \leq x \mid \mathcal{F}_{j-1})\mathbb{P}(R_t \leq y \mid \mathcal{F}_{j-1}).$$

**Intuition.** Assumption A.5 states that the current $p$-value $p_j$ and the future total discovery count $R_t$ are negatively correlated. This is intuitive: a small $p_j$ leads to a rejection at step $j$, increasing the wealth and facilitating more future discoveries (larger $R_t$). Thus, $p_j$ and $R_t$ tend to move in opposite directions. This assumption aligns with the insight of (Fisher, 2024), which similarly leverages local dependence plus monotone wealth updates to argue that early discoveries shift future thresholds upward and thereby induce the same negative association between a current $p$-value and downstream rejections.

By Lehmann's Lemma (Lehmann, 2011), if $(X, Y)$ are NQD, then $\mathrm{Cov}(f(X), g(Y)) \leq 0$ for any pair of non-decreasing functions $f$ and $g$. Equivalently, if $f$ is non-increasing and $g$ is non-decreasing (or vice-versa), then $\mathrm{Cov}(f(X), g(Y)) \geq 0$.

Under these conditions, the same covariance-decomposition technique used for $e$-values applies to $p$-values, yielding tight FDR bounds. We illustrate this for two canonical estimator types.

Consider first the LORD-type FDP estimator $\widehat{\mathrm{FDP}}(t) = \sum_{j=1}^{t} \alpha_j / (R_t \vee 1)$ commonly used in $p$-value-based LORD procedures.

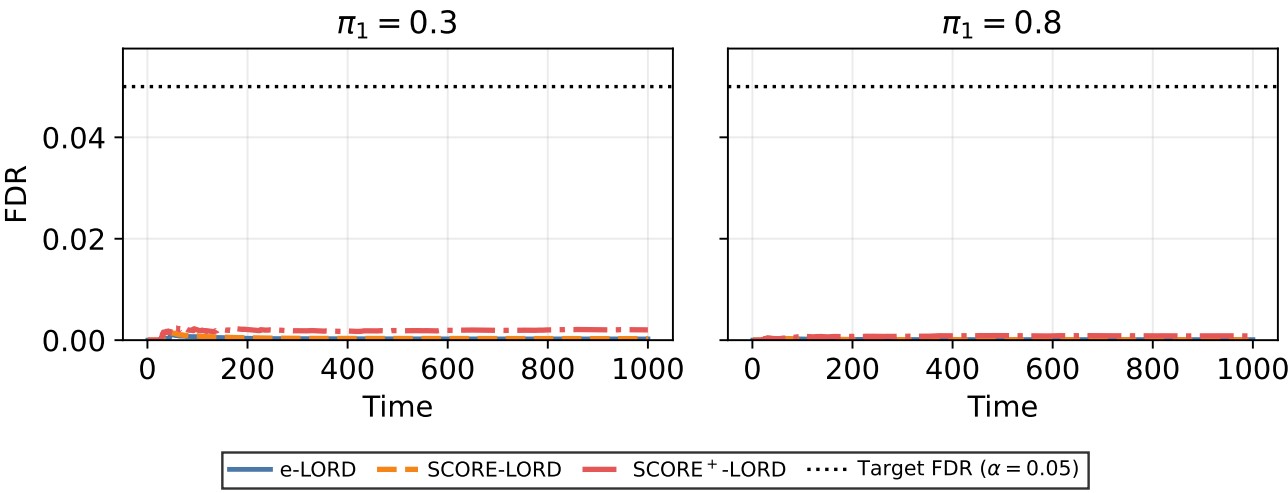

*Figure 5.* Independent setting, LORD family: FDR trajectories.

**Theorem A.6.** *Assume each null p-value is conditionally super-uniform. Under Assumption A.5, we have*

$$\mathrm{FDR}(t) \leq \mathbb{E}\left[\frac{1}{R_t \vee 1}\sum_{j=1}^{t}\alpha_j\right].$$

For SAFFRON-style procedures with candidate screening, we define the adjusted cost $C_j^{\mathrm{SAF}} = \frac{\alpha_j \mathbb{I}\{p_j > \lambda_j\}}{1-\lambda_j}$.

**Theorem A.7.** *Assume each null p-value is conditionally super-uniform. Under Assumption A.5, we have*

$$\mathrm{FDR}(t) \leq \mathbb{E}\left[\frac{1}{R_t \vee 1}\sum_{j=1}^{t}C_j^{\mathrm{SAF}}\right].$$

Finally, we verify that the same independence and monotonicity conditions suffice for $p$-value-based methods.

**Proposition A.8.** *Suppose the p-values $\{p_j\}_{j=1}^{t}$ are mutually independent. If the significance levels $\alpha_k$ are coordinate-wise non-decreasing functions of the past decisions $(\delta_1, \ldots, \delta_{k-1})$, then Assumption A.5 holds.*

Consequently, the result above shows that the familiar independence-plus-monotonicity conditions used in prior $p$-value-based analyses are sufficient to imply Assumption A.5, and hence to justify the validity of standard $p$-value procedures (e.g., LORD and SAFFRON) within our covariance-based framework. Crucially, however, our CNQD assumption is strictly weaker than full independence: independence and monotonicity merely provide a convenient sufficient condition.

As we illustrate in Sections 6 and E, synthetic and real-data experiments indicate these procedures continue to control FDR even when independence is evidently violated. This demonstrates that our theoretical conditions substantially broaden the applicable settings and improve robustness in practice.

Our analysis thereby broadens the theoretical scope of existing $p$-value-based methods. This provides theoretical reassurance for practical use: in many real-world problems independence cannot be guaranteed, but with well-designed test statistics the assumption—whose logic aligns with the alpha-investing intuition—is often easy to meet in practice.

## B. Additional Robustness Checks

### B.1. FDR Trajectories for the Main Simulations

Figures 5–8 report the FDR trajectories corresponding to the power comparisons in Section 6.

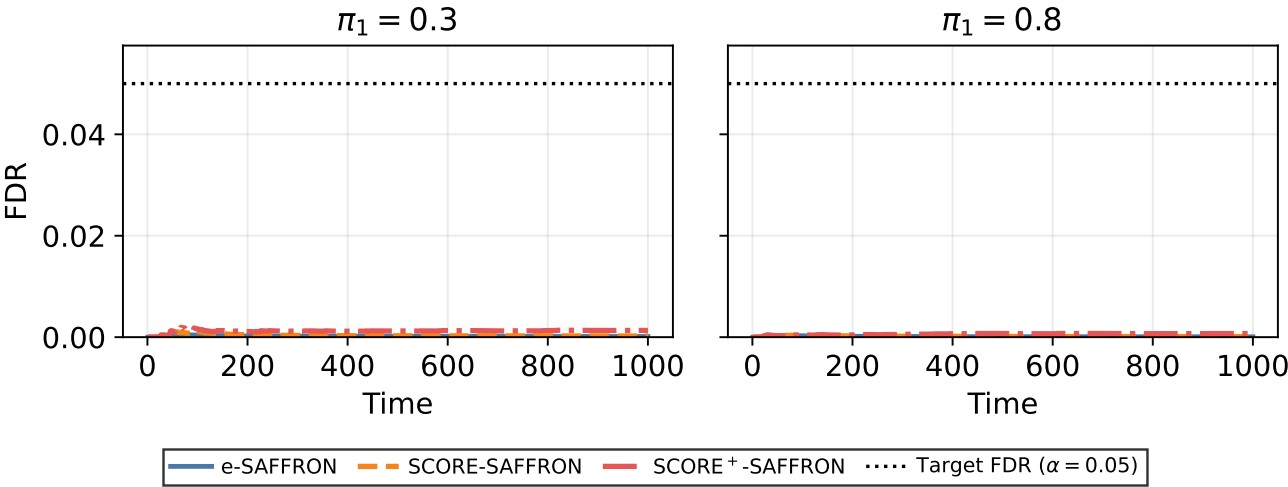

*Figure 6.* Independent setting, SAFFRON family: FDR trajectories.

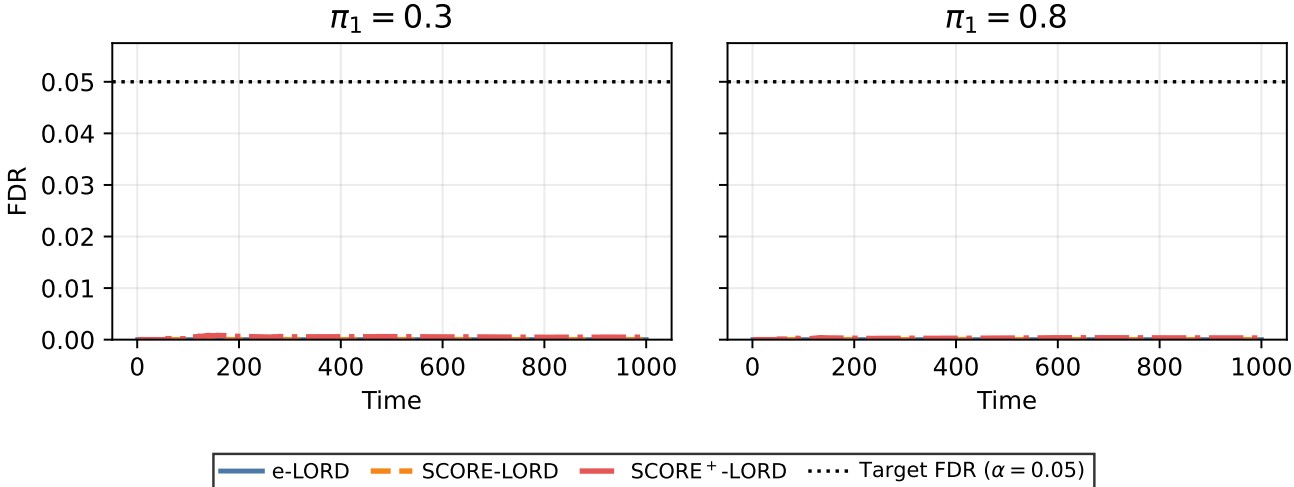

*Figure 7.* Dependent setting, LORD family: FDR trajectories.

## B.2. A Stress Test for CPQD

The SCORE$^+$ procedures use the global denominator $R_t \vee 1$ and therefore require Assumption 5.1. To illustrate the role of this assumption, we construct a deliberately adversarial data stream that induces negative association between a null $e$-value and the final number of discoveries.

We set $T = 1000$ and $\alpha = 0.05$. The truth sequence is fixed in advance: in the one-indexed notation of the paper, hypotheses $H_1, H_3, \ldots$ are non-null and $H_2, H_4, \ldots$ are null. At time $t$, let $\alpha_t$ denote the current testing level that the corresponding SCORE$^+$ procedure would use. We generate a raw observation $X_t$ and convert it to an $e$-value by

$$e_t = \frac{\mathbb{I}\{X_t \leq \alpha_t\}}{\alpha_t}.$$

Under a null hypothesis, $X_t \sim \text{Unif}(0, 1)$, so $\mathbb{E}(e_t \mid \mathcal{F}_{t-1}) = 1$. Under a non-null hypothesis, before any null trap is triggered, we set $X_t = \alpha_t/2$, producing a strong signal. Once a null step happens to generate $X_t \leq \alpha_t$ and hence a large null $e$-value, the future non-null observations are changed to $X_t = (1 + \alpha_t)/2$, eliminating subsequent alternative signals. Thus a large null $e$-value suppresses future discoveries, reversing the usual alpha-investing intuition behind CPQD.

It is important to emphasize that the setup constructed above is a highly contrived scenario. We test numerous other settings but all fail to break the FDR control. While this simulation delineates the theoretical boundaries of SCORE$^+$, it also

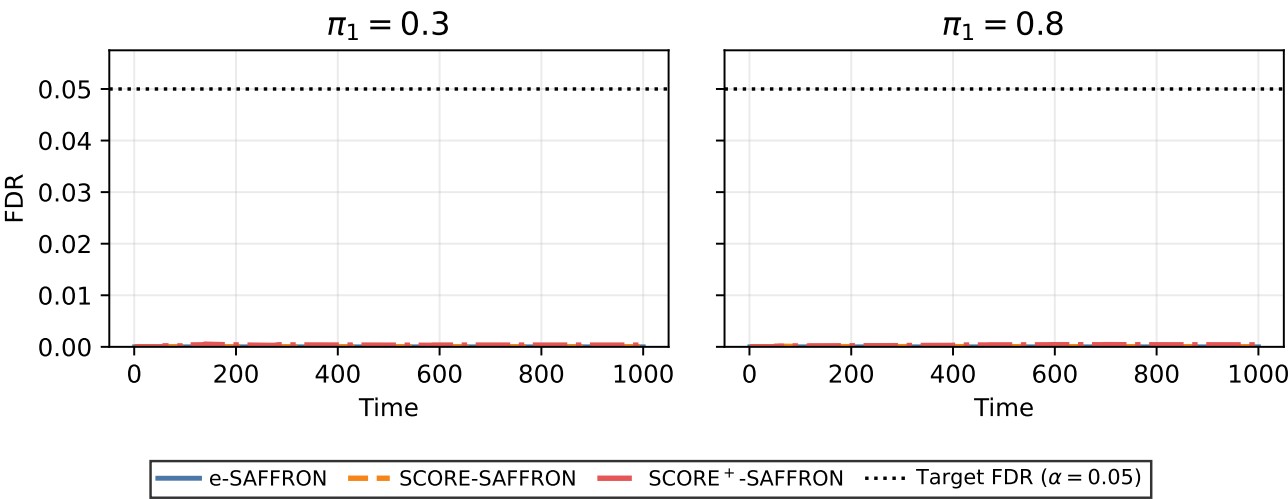

*Figure 8.* Dependent setting, SAFFRON family: FDR trajectories.

highlights that the CPQD assumption is mild, and the SCORE$^+$ procedure remains robust in general settings.

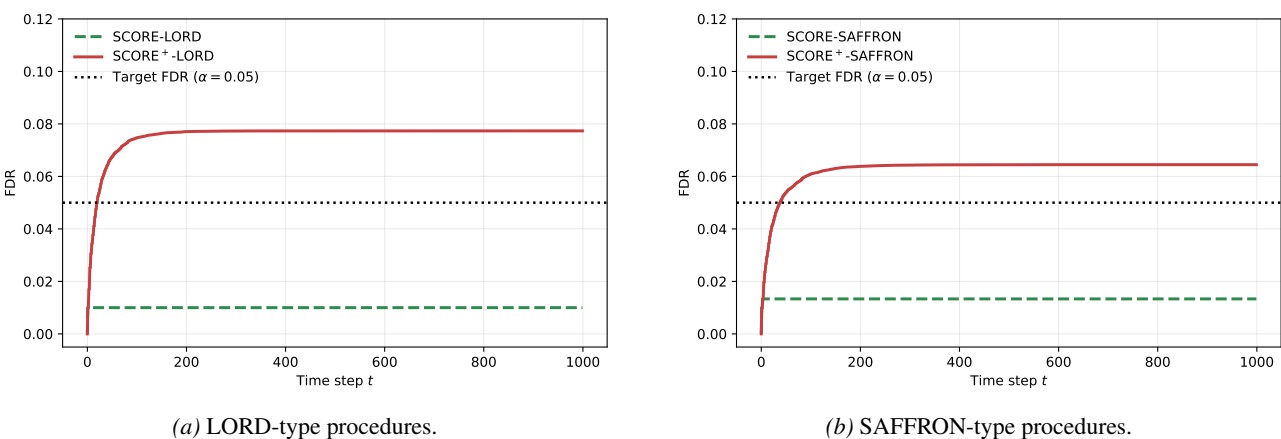

*(a)* LORD-type procedures.  *(b)* SAFFRON-type procedures.

*Figure 9.* Stress test under a deliberately CPQD-violating dependence structure. SCORE remains controlled because it does not use retroactive denominators, whereas SCORE$^+$ can exceed the target level when the positive-association mechanism is reversed.

### B.3. Sensitivity to the Target FDR Level

We also repeated the main synthetic experiments over several nominal levels $\alpha \in \{0.05, 0.10, 0.15, 0.20\}$. Tables 2 and 3 report the final-time FDR, power, and power ratios at $T = 1000$. The same ordering persists across target levels: SCORE improves over its corresponding baseline, and SCORE$^+$ gives the largest power when its dependence assumption is satisfied.

## C. Extension to Decaying-Memory FDR

The overshoot-refund idea is not tied to the ordinary, undiscounted online FDR criterion. It can also be combined with long-horizon criteria that discount old decisions. For a discount factor $d \in (0, 1]$, define the decayed numbers of false and total discoveries by

$$V_d(t) = \sum_{j \in \mathcal{H}_0(t)} d^{t-j}\delta_j, \qquad R_d(t) = \sum_{j=1}^{t} d^{t-j}\delta_j.$$

*Table 2.* Sensitivity to target FDR levels in the independent setting. Results are averaged over 500 repetitions. Ratios are computed relative to the corresponding baseline method within each family.

| $\alpha$ | $\pi_1$ | Family | FDR($T$) | | | Power($T$) | | | Power ratio | |
|---|---|---|---|---|---|---|---|---|---|---|
| | | | Baseline | SCORE | SCORE$^+$ | Baseline | SCORE | SCORE$^+$ | SCORE | SCORE$^+$ |
| 0.05 | 0.3 | LORD | 0.0003 | 0.0004 | 0.0016 | 0.0818 | 0.0886 | 0.4244 | 1.08 | 5.19 |
| 0.05 | 0.3 | SAFFRON | 0.0002 | 0.0001 | 0.0010 | 0.0981 | 0.1207 | 0.4052 | 1.23 | 4.13 |
| 0.05 | 0.8 | LORD | 0.0000 | 0.0001 | 0.0008 | 0.0916 | 0.1256 | 0.5099 | 1.37 | 5.56 |
| 0.05 | 0.8 | SAFFRON | 0.0001 | 0.0001 | 0.0007 | 0.1973 | 0.2479 | 0.4986 | 1.26 | 2.53 |
| 0.10 | 0.3 | LORD | 0.0006 | 0.0006 | 0.0044 | 0.0881 | 0.0978 | 0.4678 | 1.11 | 5.31 |
| 0.10 | 0.3 | SAFFRON | 0.0005 | 0.0003 | 0.0027 | 0.1084 | 0.1346 | 0.4463 | 1.24 | 4.12 |
| 0.10 | 0.8 | LORD | 0.0001 | 0.0002 | 0.0020 | 0.0993 | 0.1424 | 0.5605 | 1.43 | 5.64 |
| 0.10 | 0.8 | SAFFRON | 0.0001 | 0.0001 | 0.0015 | 0.2162 | 0.2707 | 0.5434 | 1.25 | 2.51 |
| 0.15 | 0.3 | LORD | 0.0005 | 0.0006 | 0.0058 | 0.0947 | 0.1042 | 0.4919 | 1.10 | 5.19 |
| 0.15 | 0.3 | SAFFRON | 0.0004 | 0.0004 | 0.0038 | 0.1145 | 0.1410 | 0.4691 | 1.23 | 4.10 |
| 0.15 | 0.8 | LORD | 0.0001 | 0.0003 | 0.0033 | 0.1059 | 0.1552 | 0.5907 | 1.47 | 5.58 |
| 0.15 | 0.8 | SAFFRON | 0.0001 | 0.0002 | 0.0023 | 0.2278 | 0.2854 | 0.5702 | 1.25 | 2.50 |
| 0.20 | 0.3 | LORD | 0.0013 | 0.0016 | 0.0088 | 0.0973 | 0.1077 | 0.5113 | 1.11 | 5.26 |
| 0.20 | 0.3 | SAFFRON | 0.0008 | 0.0008 | 0.0051 | 0.1180 | 0.1457 | 0.4868 | 1.23 | 4.13 |
| 0.20 | 0.8 | LORD | 0.0002 | 0.0004 | 0.0046 | 0.1098 | 0.1636 | 0.6154 | 1.49 | 5.60 |
| 0.20 | 0.8 | SAFFRON | 0.0002 | 0.0003 | 0.0032 | 0.2366 | 0.2964 | 0.5914 | 1.25 | 2.50 |

Following the decaying-memory FDR criterion of Ramdas et al. (2017), define

$$\text{mem-FDR}_d(t) = \mathbb{E}\left[\frac{V_d(t)}{R_d(t) \vee 1}\right],$$

which we use here to illustrate how the SCORE accounting extends to memory-based objectives. Since $R_d(t)$ is discounted and may lie in $(0, 1)$, the denominator $R_d(t) \vee 1$ can be conservative.

Let $O_j = (\alpha_j e_j - 1)_+$. For LORD-type procedures the SCORE-adjusted cost is

$$C_j^{\text{S-LORD}} = (\alpha_j - O_j)_+,$$

whereas for SAFFRON-type procedures with predictable candidate thresholds $\lambda_j \in (0, 1)$ it is

$$C_j^{\text{S-SAF}} = \left(\frac{\alpha_j(1 - \lambda_j e_j)}{1 - \lambda_j} - O_j\right)_+.$$

This gives two memory analogues, matching the SCORE and SCORE$^+$ procedures in the main text.

**Retroactive Update.** The SCORE$^+$ analogue uses the same retroactive denominator as the target mem-FDP. This gives the LORD and SAFFRON estimators

$$\widehat{\text{mem-FDP}}_{d,\text{S}^+\text{-LORD}}(t) = \frac{\sum_{j=1}^t d^{t-j} C_j^{\text{S-LORD}}}{R_d(t) \vee 1}, \qquad \widehat{\text{mem-FDP}}_{d,\text{S}^+\text{-SAF}}(t) = \frac{\sum_{j=1}^t d^{t-j} C_j^{\text{S-SAF}}}{R_d(t) \vee 1}.$$

These estimators are the memory versions of SCORE$^+$-LORD and SCORE$^+$-SAFFRON: costs, discoveries, and overshoot refunds are all discounted by the same factor $d^{t-j}$, and past costs are re-evaluated against the current decayed discovery total. As in the undiscounted SCORE$^+$ case, the denominator is not predictable at time $j$; a validity proof therefore requires a memory analogue of CPQD, with $R_t$ replaced by $R_d(t)$.

**Predictable Update.** The basic SCORE analogue avoids the retroactive denominator by charging each test against a predictable denominator. The subtle point is that the previous discoveries must first be put on the correct time scale. Since

$$R_d(j-1) = \sum_{k=1}^{j-1} d^{j-1-k} \delta_k,$$

*Table 3.* Sensitivity to target FDR levels in the dependent setting. Results are averaged over 500 repetitions. Ratios are computed relative to the corresponding baseline method within each family.

| $\alpha$ | $\pi_1$ | Family | FDR($T$) | | | Power($T$) | | | Power ratio | |
| --- | --- | --- | --- | --- | --- | --- | --- | --- | --- | --- |
| | | | Baseline | SCORE | SCORE$^+$ | Baseline | SCORE | SCORE$^+$ | SCORE | SCORE$^+$ |
| 0.05 | 0.3 | LORD | 0.0000 | 0.0000 | 0.0005 | 0.0706 | 0.0760 | 0.3239 | 1.08 | 4.59 |
| 0.05 | 0.3 | SAFFRON | 0.0000 | 0.0000 | 0.0004 | 0.0894 | 0.1431 | 0.3212 | 1.60 | 3.59 |
| 0.05 | 0.8 | LORD | 0.0000 | 0.0000 | 0.0003 | 0.0827 | 0.1043 | 0.3862 | 1.26 | 4.67 |
| 0.05 | 0.8 | SAFFRON | 0.0000 | 0.0000 | 0.0004 | 0.1744 | 0.2362 | 0.3925 | 1.35 | 2.25 |
| 0.10 | 0.3 | LORD | 0.0001 | 0.0001 | 0.0018 | 0.0756 | 0.0817 | 0.3546 | 1.08 | 4.69 |
| 0.10 | 0.3 | SAFFRON | 0.0000 | 0.0000 | 0.0014 | 0.0965 | 0.1546 | 0.3502 | 1.60 | 3.63 |
| 0.10 | 0.8 | LORD | 0.0000 | 0.0000 | 0.0013 | 0.0890 | 0.1145 | 0.4312 | 1.29 | 4.84 |
| 0.10 | 0.8 | SAFFRON | 0.0000 | 0.0001 | 0.0014 | 0.1858 | 0.2531 | 0.4356 | 1.36 | 2.34 |
| 0.15 | 0.3 | LORD | 0.0001 | 0.0001 | 0.0030 | 0.0801 | 0.0865 | 0.3726 | 1.08 | 4.65 |
| 0.15 | 0.3 | SAFFRON | 0.0000 | 0.0002 | 0.0024 | 0.1007 | 0.1620 | 0.3669 | 1.61 | 3.64 |
| 0.15 | 0.8 | LORD | 0.0001 | 0.0001 | 0.0026 | 0.0947 | 0.1237 | 0.4647 | 1.31 | 4.91 |
| 0.15 | 0.8 | SAFFRON | 0.0000 | 0.0001 | 0.0028 | 0.1947 | 0.2655 | 0.4661 | 1.36 | 2.39 |
| 0.20 | 0.3 | LORD | 0.0004 | 0.0004 | 0.0055 | 0.0846 | 0.0920 | 0.3944 | 1.09 | 4.66 |
| 0.20 | 0.3 | SAFFRON | 0.0003 | 0.0002 | 0.0038 | 0.1069 | 0.1706 | 0.3880 | 1.60 | 3.63 |
| 0.20 | 0.8 | LORD | 0.0001 | 0.0002 | 0.0046 | 0.0969 | 0.1279 | 0.4929 | 1.32 | 5.09 |
| 0.20 | 0.8 | SAFFRON | 0.0001 | 0.0002 | 0.0043 | 0.1990 | 0.2729 | 0.4913 | 1.37 | 2.47 |

the decayed number of past discoveries immediately before testing $H_j$ is $dR_d(j-1)$, not $R_d(j-1)$. If $H_j$ is rejected, the current discovery and all previous discoveries contribute at least

$$d^{t-j}\{1 + dR_d(j-1)\}$$

to $R_d(t)$ at time $t$. The predictable conservative denominator for the $j$th charge at time $t$ is therefore

$$D_{j,t}^{\text{SCORE}} = d^{t-j}\{1 + dR_d(j-1)\}.$$

This denominator is strictly positive because $d > 0$; no additional "$\vee 1$" correction is needed. Indeed, on the event $\delta_j = 1$, we have $R_d(t) \vee 1 \geq D_{j,t}^{\text{SCORE}}$, so each false discovery contribution satisfies

$$\frac{d^{t-j}\delta_j}{R_d(t) \vee 1} \leq \frac{d^{t-j}\delta_j}{D_{j,t}^{\text{SCORE}}}.$$

This leads to the memory versions of SCORE-LORD and SCORE-SAFFRON:

$$\widehat{\text{mem-FDP}}_{d,\text{S-LORD}}(t) = \sum_{j=1}^{t} \frac{d^{t-j}C_j^{\text{S-LORD}}}{D_{j,t}^{\text{SCORE}}}, \qquad \widehat{\text{mem-FDP}}_{d,\text{S-SAF}}(t) = \sum_{j=1}^{t} \frac{d^{t-j}C_j^{\text{S-SAF}}}{D_{j,t}^{\text{SCORE}}}.$$

These two estimators reduce to the ordinary SCORE-LORD and SCORE-SAFFRON estimators when $d = 1$. They are more conservative than the SCORE$^+$ memory estimators, but their denominators are predictable at the time each charge is incurred. Thus they follow the same proof strategy as the basic SCORE procedures and do not require CPQD.

## D. Proofs

### D.1. Proof of Lemma 3.1

*Proof.* If $y < 1$, then $\mathbb{I}(y \geq 1) = 0$ and $y - (y-1)_+ = y - 0 = y \geq 0$. If $y \geq 1$, then $\mathbb{I}(y \geq 1) = 1$ and $y - (y-1)_+ = y - (y-1) = 1$. In both cases, the inequality holds (with equality for $y \geq 1$). $\square$

## D.2. Proof of Theorem 3.2

*Proof.* Using (4) and the assumption on $C_j$:

$$\begin{aligned}
\mathbb{E}[\delta_j \mid \mathcal{F}_{j-1}] &\leq \mathbb{E}[\alpha_j e_j - O_j \mid \mathcal{F}_{j-1}] \\
&= \mathbb{E}[\alpha_j e_j \mid \mathcal{F}_{j-1}] - \mathbb{E}[O_j \mid \mathcal{F}_{j-1}] \\
&\leq \mathbb{E}[C_j \mid \mathcal{F}_{j-1}] - \mathbb{E}[O_j \mid \mathcal{F}_{j-1}] \\
&= \mathbb{E}[C_j - O_j \mid \mathcal{F}_{j-1}] \\
&\leq \mathbb{E}[(C_j - O_j)_+ \mid \mathcal{F}_{j-1}].
\end{aligned}$$

Then we can use this inequality to bound the online FDR:

$$\begin{aligned}
\mathrm{FDR}(t) = \mathbb{E}\left[\frac{\sum_{j\in\mathcal{H}_0(t)}\delta_j}{R_t \vee 1}\right] &\leq \mathbb{E}\left[\sum_{j\in\mathcal{H}_0(t)}\frac{\delta_j}{R_{j-1}+1}\right] \quad \text{(since } R_{j-1}+1 \leq (R_t \vee 1) \text{ for every } j \in \{j \leq t : \delta_j = 1\}) \\
&= \sum_{j\in\mathcal{H}_0(t)}\mathbb{E}\left[\frac{\mathbb{E}[\delta_j \mid \mathcal{F}_{j-1}]}{R_{j-1}+1}\right] \\
&\leq \sum_{j\in\mathcal{H}_0(t)}\mathbb{E}\left[\frac{\mathbb{E}[(C_j-O_j)_+ \mid \mathcal{F}_{j-1}]}{R_{j-1}+1}\right] \quad \text{(by the inequality above)} \\
&= \mathbb{E}\left[\sum_{j\in\mathcal{H}_0(t)}\frac{(C_j-O_j)_+}{R_{j-1}+1}\right] \\
&\leq \mathbb{E}\left[\sum_{j=1}^{t}\frac{(C_j-O_j)_+}{R_{j-1}+1}\right].
\end{aligned}$$

The last term is precisely $\mathbb{E}[\widehat{\mathrm{FDP}}_{\text{SCORE}}(t)]$. $\qquad\square$

## D.3. Proof of Proposition 4.1

*Proof.* We need to show that $\sum_{j=1}^{t}\frac{C_j^{\text{S-LOND}}}{R_{j-1}+1} \leq \alpha$, where $C_j^{\text{S-LOND}} = (\alpha_j^{\text{S-LOND}} - O_j)_+ = \alpha_j^{\text{S-LOND}} - \min(\alpha_j^{\text{S-LOND}}, O_j)$. Let $Q_t = \alpha + \sum_{k=1}^{t-1}\frac{\min(O_k,\alpha_k^{\text{S-LOND}})}{R_{k-1}+1}$. The update rule is $\alpha_t^{\text{S-LOND}} = \gamma_t(R_{t-1}+1)Q_t$. Substituting this into the sum:

$$\begin{aligned}
\sum_{j=1}^{t}\frac{\alpha_j^{\text{S-LOND}} - \min(\alpha_j^{\text{S-LOND}},O_j)}{R_{j-1}+1} &= \sum_{j=1}^{t}\left(\frac{\alpha_j^{\text{S-LOND}}}{R_{j-1}+1} - \frac{\min(\alpha_j^{\text{S-LOND}},O_j)}{R_{j-1}+1}\right) \\
&= \sum_{j=1}^{t}(\gamma_j Q_j - (Q_{j+1} - Q_j)) \\
&= \sum_{j=1}^{t}\gamma_j Q_j - (Q_{t+1} - Q_1).
\end{aligned}$$

Since $Q_j$ is non-decreasing (as the refund term is non-negative), we have $Q_j \leq Q_t \leq Q_{t+1}$. Also $\sum_{j=1}^{t}\gamma_j \leq \sum_{j=1}^{\infty}\gamma_j = 1$. Thus $\sum_{j=1}^{t}\gamma_j Q_j \leq Q_{t+1}\sum_{j=1}^{t}\gamma_j \leq Q_{t+1}$. Therefore, the total sum is $\leq Q_{t+1} - Q_{t+1} + Q_1 = Q_1 = \alpha$. $\qquad\square$

## D.4. Proof of Proposition 4.2

*Proof.* We prove $\alpha_t^{\text{S-LOND}} \geq \alpha_t^{\text{e-LOND}}$ by induction on $t$.

*Base Case ($t = 1$):* Both algorithms start with $R_0 = 0$ and no prior refunds.

$$\alpha_1^{\text{S-LOND}} = \gamma_1(0+1)(\alpha+0) = \gamma_1\alpha = \alpha_1^{\text{e-LOND}}.$$

Thus, $\delta_1^{\text{S-LOND}} = \delta_1^{\text{e-LOND}}$, implying $R_1^{\text{S-LOND}} = R_1^{\text{e-LOND}}$.

*Inductive Step:* Assume that for all $j < t$, we have $\alpha_j^{\text{S-LOND}} \geq \alpha_j^{\text{e-LOND}}$ and $R_j^{\text{S-LOND}} \geq R_j^{\text{e-LOND}}$. Consider the update at time $t$:

$$\alpha_t^{\text{S-LOND}} = \gamma_t(R_{t-1}^{\text{S-LOND}} + 1)\left(\alpha + \sum_{k=1}^{t-1} \frac{\min(O_k, \alpha_k^{\text{S-LOND}})}{R_{k-1}^{\text{S-LOND}} + 1}\right).$$

Since the refund term (summation) is non-negative and $R_{t-1}^{\text{S-LOND}} \geq R_{t-1}^{\text{e-LOND}}$ (by inductive hypothesis), we have:

$$\alpha_t^{\text{S-LOND}} \geq \gamma_t(R_{t-1}^{\text{e-LOND}} + 1)\alpha = \alpha_t^{\text{e-LOND}}.$$

Now consider the decision at time $t$. If e-LOND rejects $H_t$, then $e_t \geq 1/\alpha_t^{\text{e-LOND}}$. Since $\alpha_t^{\text{S-LOND}} \geq \alpha_t^{\text{e-LOND}}$, then $e_t \geq 1/\alpha_t^{\text{S-LOND}}$. Thus SCORE-LOND also rejects $H_t$. Therefore, $\delta_t^{\text{S-LOND}} \geq \delta_t^{\text{e-LOND}}$, which implies $R_t^{\text{S-LOND}} \geq R_t^{\text{e-LOND}}$. $\qquad\square$

### D.5. Proof of Proposition 4.3

*Proof.* We first prove the validity of the update rule, then establish the dominance over e-LORD.

**Part 1: Validity.** Let $W_t = \alpha - \sum_{j=1}^{t-1} \frac{C_j^{\text{S-LORD}}}{R_{j-1}+1}$, where $C_j^{\text{S-LORD}} = (\alpha_j^{\text{S-LORD}} - O_j)_+$. We aim to show $W_{t+1} \geq 0$ for all $t$. The update rule is $\alpha_t^{\text{S-LORD}} = \omega_t(R_{t-1} + 1)W_t$. We prove $W_t \geq 0$ for all $t$ by induction.

*Base case:* $W_1 = \alpha > 0$.

*Inductive step:* Assume $W_t \geq 0$. Then $\alpha_t^{\text{S-LORD}} = \omega_t(R_{t-1} + 1)W_t \geq 0$. Since $C_t^{\text{S-LORD}} = (\alpha_t^{\text{S-LORD}} - O_t)_+ \leq \alpha_t^{\text{S-LORD}}$, we have

$$W_{t+1} = W_t - \frac{C_t^{\text{S-LORD}}}{R_{t-1} + 1} \geq W_t - \frac{\alpha_t^{\text{S-LORD}}}{R_{t-1} + 1} = W_t(1 - \omega_t) \geq 0,$$

where the last inequality follows from $W_t \geq 0$ and $\omega_t \in (0, 1)$.

By mathematical induction, $W_t \geq 0$ for all $t$. Hence,

$$\sum_{j=1}^{t} \frac{C_j^{\text{S-LORD}}}{R_{j-1} + 1} = \alpha - W_{t+1} \leq \alpha.$$

**Part 2: Dominance over e-LORD.**

We define the "wealth" $W_t$ for both algorithms. For e-LORD:

$$W_t^{\text{LORD}} = \alpha - \sum_{j=1}^{t-1} \frac{\alpha_j^{\text{LORD}}}{R_{j-1}^{\text{LORD}} + 1}.$$

From the definition of e-LORD, we have $\frac{\alpha_j^{\text{LORD}}}{R_{j-1}^{\text{LORD}}+1} = \omega_j W_j^{\text{LORD}}$. Thus, the wealth evolves as:

$$W_{t+1}^{\text{LORD}} = W_t^{\text{LORD}} - \omega_t W_t^{\text{LORD}} = W_t^{\text{LORD}}(1 - \omega_t).$$

Solving this recurrence yields $W_t^{\text{LORD}} = \alpha \prod_{j=1}^{t-1}(1 - \omega_j)$.

For SCORE-LORD:

$$W_t^{\text{SCORE}} = \alpha - \sum_{j=1}^{t-1} \frac{C_j^{\text{SCORE}}}{R_{j-1}^{\text{SCORE}} + 1}, \quad \text{where } C_j^{\text{SCORE}} = (\alpha_j^{\text{SCORE}} - O_j)_+.$$

The update rule implies $\frac{\alpha_t^{\text{SCORE}}}{R_{t-1}^{\text{SCORE}}+1} = \omega_t W_t^{\text{SCORE}}$. The wealth update is:

$$W_{t+1}^{\text{SCORE}} = W_t^{\text{SCORE}} - \frac{C_t^{\text{SCORE}}}{R_{t-1}^{\text{SCORE}} + 1}.$$

We consider two cases for $C_t^{\text{SCORE}}$:

- Case 1: $\alpha_t^{\text{SCORE}} \geq O_t$. Then $C_t^{\text{SCORE}} = \alpha_t^{\text{SCORE}} - O_t$.

$$W_{t+1}^{\text{SCORE}} = W_t^{\text{SCORE}}(1 - \omega_t) + \frac{O_t}{R_{t-1}^{\text{SCORE}} + 1}.$$

- Case 2: $\alpha_t^{\text{SCORE}} < O_t$. Then $C_t^{\text{SCORE}} = 0$.

$$W_{t+1}^{\text{SCORE}} = W_t^{\text{SCORE}}.$$

In both cases, we have $W_{t+1}^{\text{SCORE}} \geq W_t^{\text{SCORE}}(1 - \omega_t)$. Moreover, if $O_t > 0$, then $W_{t+1}^{\text{SCORE}} > W_t^{\text{SCORE}}(1 - \omega_t)$. This is because in Case 1, the extra term $\frac{O_t}{R_{t-1}^{\text{SCORE}}+1} > 0$; in Case 2, $W_t^{\text{SCORE}} > W_t^{\text{SCORE}}(1 - \omega_t)$ since $\omega_t \in (0, 1)$.

We now prove the dominance results formally.

1. **Wealth Dominance:** We claim $W_t^{\text{SCORE}} \geq W_t^{\text{LORD}}$ for all $t$, with strict inequality if $\exists j < t$ such that $O_j > 0$.

   - Base case $t = 1$: $W_1^{\text{SCORE}} = W_1^{\text{LORD}} = \alpha$.
   - Inductive step: Assume $W_t^{\text{SCORE}} \geq W_t^{\text{LORD}}$. From the recurrence derived above:

   $$W_{t+1}^{\text{SCORE}} \geq W_t^{\text{SCORE}}(1 - \omega_t) \geq W_t^{\text{LORD}}(1 - \omega_t) = W_{t+1}^{\text{LORD}}.$$

   - Strictness: If $O_t > 0$, we showed strictly $W_{t+1}^{\text{SCORE}} > W_t^{\text{SCORE}}(1 - \omega_t) \geq W_{t+1}^{\text{LORD}}$. This gap propagates because the update factor $1 - \omega_k > 0$ preserves strict inequality. Thus, $W_t^{\text{SCORE}} > W_t^{\text{LORD}}$ for all $t > j$, where $j$ is the index of the first overshoot.

2. **Rejection Count Dominance:** We claim $R_t^{\text{SCORE}} \geq R_t^{\text{LORD}}$ for all $t$.

   - Base case $t = 0$: $R_0^{\text{SCORE}} = R_0^{\text{LORD}} = 0$.
   - Inductive step: Assume $R_t^{\text{SCORE}} \geq R_t^{\text{LORD}}$. From the Wealth Dominance, we have $W_{t+1}^{\text{SCORE}} \geq W_{t+1}^{\text{LORD}}$. The significance level for the next step is determined by $\alpha_{t+1} = \omega_{t+1}(R_t + 1)W_{t+1}$. Since both $R_t^{\text{SCORE}} \geq R_t^{\text{LORD}}$ and $W_{t+1}^{\text{SCORE}} \geq W_{t+1}^{\text{LORD}}$, it follows that $\alpha_{t+1}^{\text{SCORE}} \geq \alpha_{t+1}^{\text{LORD}}$. Consequently, the rejection threshold for SCORE is lower, i.e., $1/\alpha_{t+1}^{\text{SCORE}} \leq 1/\alpha_{t+1}^{\text{LORD}}$. This implies that if e-LORD rejects ($e_{t+1} \geq 1/\alpha_{t+1}^{\text{LORD}}$), then SCORE must also reject, so $\delta_{t+1}^{\text{SCORE}} \geq \delta_{t+1}^{\text{LORD}}$. Finally, since $R_{t+1} = R_t + \delta_{t+1}$ and both terms on the right are greater for SCORE, we conclude $R_{t+1}^{\text{SCORE}} \geq R_{t+1}^{\text{LORD}}$.

3. **Threshold Dominance:** Combining the above,

$$\alpha_t^{\text{SCORE}} = \omega_t(R_{t-1}^{\text{SCORE}} + 1)W_t^{\text{SCORE}}$$
$$\geq \omega_t(R_{t-1}^{\text{LORD}} + 1)W_t^{\text{LORD}} = \alpha_t^{\text{LORD}}.$$

The inequality is strict if either wealth is strictly larger ($W_t^{\text{SCORE}} > W_t^{\text{LORD}}$) or the rejection count is strictly larger. Since a strict increase in wealth is triggered by any prior overshoot $O_j > 0$ (i.e., $e_j > 1/\alpha_j^{\text{SCORE}}$), this is the sufficient condition for $\alpha_t^{\text{SCORE}} > \alpha_t^{\text{LORD}}$ for all subsequent $t$.

$\square$

### D.6. Proof of Proposition 4.4

*Proof.* We first prove the validity of the update rule, then establish the dominance over e-SAFFRON.

**Part 1: Validity.** Let $W_t = \alpha - \sum_{j=1}^{t-1} \frac{C_j^{\text{S-SAF}}}{R_{j-1}+1}$. The update rule is $\alpha_t^{\text{S-SAF}} = \omega_t(1 - \lambda_t)(R_{t-1} + 1)W_t$. For the cost at step $t$, note that

$$C_t^{\text{S-SAF}} = \left( \frac{\alpha_t^{\text{S-SAF}}(1 - \lambda_t e_t)}{1 - \lambda_t} - O_t \right)_+ \leq \left( \frac{\alpha_t^{\text{S-SAF}}(1 - \lambda_t e_t)}{1 - \lambda_t} \right)_+ \leq \frac{\alpha_t^{\text{S-SAF}}}{1 - \lambda_t}.$$

Therefore,

$$W_{t+1} = W_t - \frac{C_t^{\text{S-SAF}}}{R_{t-1} + 1} \geq W_t - \frac{\alpha_t^{\text{S-SAF}}}{(1 - \lambda_t)(R_{t-1} + 1)}.$$

Using the update rule $\alpha_t^{\text{S-SAF}} = \omega_t(1 - \lambda_t)(R_{t-1} + 1)W_t$, we have

$$\frac{\alpha_t^{\text{S-SAF}}}{(1 - \lambda_t)(R_{t-1} + 1)} = \omega_t W_t,$$

and hence

$$W_{t+1} \geq W_t - \omega_t W_t = W_t(1 - \omega_t).$$

Since $W_1 = \alpha > 0$ and $\omega_t \in (0, 1)$, it follows by induction that $W_t > 0$ for all $t$. Thus, we have $\sum_{j=1}^{t-1} \frac{C_j^{\text{S-SAF}}}{R_{j-1}+1} \leq \alpha$ for all $t$, which implies $\widehat{\text{FDP}}_{\text{S-SAF}}(t) \leq \alpha$ almost surely.

Now it is sufficient to show that $\text{FDR}(t) \leq \mathbb{E}[\widehat{\text{FDP}}_{\text{S-SAF}}(t)]$. To see this, fix any null hypothesis $j$. By Lemma 3.1, we know that $\delta_j = \mathbb{I}(\alpha_j e_j \geq 1) \leq \alpha_j e_j - O_j$. Taking the conditional expectation given $\mathcal{F}_{j-1}$:

$$\mathbb{E}[\delta_j \mid \mathcal{F}_{j-1}] \leq \alpha_j \mathbb{E}[e_j \mid \mathcal{F}_{j-1}] - \mathbb{E}[O_j \mid \mathcal{F}_{j-1}]. \tag{9}$$

Consider the auxiliary term $X_j := \frac{\alpha_j(1 - \lambda_j e_j)}{1 - \lambda_j}$. Note that the function $f(x) = \frac{1 - \lambda_j x}{1 - \lambda_j}$ satisfies $f(x) \geq x$ for all $x \in [0, 1]$ (since $\frac{1 - \lambda_j x}{1 - \lambda_j} - x = \frac{1-x}{1-\lambda_j} \geq 0$). By the conditional e-validity $\mathbb{E}[e_j \mid \mathcal{F}_{j-1}] \leq 1$, we have:

$$\mathbb{E}[X_j \mid \mathcal{F}_{j-1}] = \frac{\alpha_j(1 - \lambda_j \mathbb{E}[e_j \mid \mathcal{F}_{j-1}])}{1 - \lambda_j} \geq \alpha_j \mathbb{E}[e_j \mid \mathcal{F}_{j-1}].$$

Substituting this back into (9):

$$\mathbb{E}[\delta_j \mid \mathcal{F}_{j-1}] \leq \mathbb{E}[X_j \mid \mathcal{F}_{j-1}] - \mathbb{E}[O_j \mid \mathcal{F}_{j-1}] = \mathbb{E}[X_j - O_j \mid \mathcal{F}_{j-1}].$$

Since $C_j^{\text{S-SAF}} = (X_j - O_j)_+ \geq X_j - O_j$, we obtain the crucial super-uniformity property:

$$\mathbb{E}[\delta_j \mid \mathcal{F}_{j-1}] \leq \mathbb{E}[C_j^{\text{S-SAF}} \mid \mathcal{F}_{j-1}].$$

Finally, following the standard GAI argument:

$$\begin{aligned}
\text{FDR}(t) = \mathbb{E}\left[\sum_{j \in \mathcal{H}_0(t)} \frac{\delta_j}{R_t \vee 1}\right] &\leq \mathbb{E}\left[\sum_{j \in \mathcal{H}_0(t)} \frac{\delta_j}{R_{j-1} + 1}\right] \\
&= \sum_{j \in \mathcal{H}_0(t)} \mathbb{E}\left[\frac{\mathbb{E}[\delta_j \mid \mathcal{F}_{j-1}]}{R_{j-1} + 1}\right] \\
&\leq \sum_{j \in \mathcal{H}_0(t)} \mathbb{E}\left[\frac{\mathbb{E}[C_j^{\text{S-SAF}} \mid \mathcal{F}_{j-1}]}{R_{j-1} + 1}\right] \\
&\leq \mathbb{E}\left[\sum_{j=1}^{t} \frac{C_j^{\text{S-SAF}}}{R_{j-1} + 1}\right] = \mathbb{E}[\widehat{\text{FDP}}_{\text{S-SAF}}(t)].
\end{aligned}$$

Since $\widehat{\text{FDP}}_{\text{S-SAF}}(t) \leq \alpha$, we have $\text{FDR}(t) \leq \alpha$.

**Part 2: Dominance over e-SAFFRON.**

*Step 1: Wealth and Threshold Recursions.* For e-SAFFRON, the wealth $W_t^{\text{SAF}}$ evolves as:

$$W_{t+1}^{\text{SAF}} = W_t^{\text{SAF}} - \frac{\alpha_t^{\text{SAF}} \mathbb{I}\{e_t < 1/\lambda_t\}}{(1 - \lambda_t)(R_{t-1}^{\text{SAF}} + 1)}.$$

Substituting $\alpha_t^{\text{SAF}} = \omega_t(1 - \lambda_t)(R_{t-1}^{\text{SAF}} + 1)W_t^{\text{SAF}}$, we get the multiplicative update:

$$W_{t+1}^{\text{SAF}} = W_t^{\text{SAF}}\left(1 - \omega_t \mathbb{I}\{e_t < 1/\lambda_t\}\right).$$

For SCORE-SAFFRON, the wealth $W_t^{\text{SCORE}}$ evolves as:

$$W_{t+1}^{\text{SCORE}} = W_t^{\text{SCORE}} - \frac{1}{R_{t-1}^{\text{SCORE}} + 1} \left( \frac{\alpha_t^{\text{SCORE}}(1 - \lambda_t e_t)}{1 - \lambda_t} - O_t \right)_+ .$$

Substituting $\alpha_t^{\text{SCORE}}$ similarly yields:

$$W_{t+1}^{\text{SCORE}} = W_t^{\text{SCORE}} - \left( \omega_t W_t^{\text{SCORE}}(1 - \lambda_t e_t) - \frac{O_t}{R_{t-1}^{\text{SCORE}} + 1} \right)_+ .$$

*Step 2: Comparison of Wealth Updates.* We claim that for all $t$, the multiplicative wealth factor for SCORE is at least that of e-SAFFRON:

$$\frac{W_{t+1}^{\text{SCORE}}}{W_t^{\text{SCORE}}} \geq \frac{W_{t+1}^{\text{SAF}}}{W_t^{\text{SAF}}}.$$

Consider the two cases for $e_t$:

- **Case 1:** $e_t \geq 1/\lambda_t$. For e-SAFFRON, $\mathbb{I}\{e_t < 1/\lambda_t\} = 0$, so the factor is 1. For SCORE, $1 - \lambda_t e_t \leq 0$. Since $O_t \geq 0$, the term inside $(\dots)_+$ is non-positive, so the cost is 0. The factor is 1. Inequality holds as equality.

- **Case 2:** $e_t < 1/\lambda_t$. For e-SAFFRON, the factor is $1 - \omega_t$. For SCORE, note that $(x - y)_+ \leq x$ for $y \geq 0$. Here $O_t \geq 0$, so

$$\left( \omega_t W_t^{\text{SCORE}}(1 - \lambda_t e_t) - \frac{O_t}{R_{t-1}^{\text{SCORE}} + 1} \right)_+ \leq \omega_t W_t^{\text{SCORE}}(1 - \lambda_t e_t).$$

    Thus,

$$W_{t+1}^{\text{SCORE}} \geq W_t^{\text{SCORE}} \left( 1 - \omega_t(1 - \lambda_t e_t) \right).$$

    Since $e_t \geq 0$, we have $0 < 1 - \lambda_t e_t \leq 1$, and thus $1 - \omega_t(1 - \lambda_t e_t) \geq 1 - \omega_t$. Inequality holds strictly if $e_t > 0$.

**Step 3: Global Dominance by Induction.** We prove $\alpha_t^{\text{SCORE}} \geq \alpha_t^{\text{SAF}}$ and $W_t^{\text{SCORE}} \geq W_t^{\text{SAF}}$ for all $t \geq 1$ by induction.

**Base case ($t = 1$):** $W_1^{\text{SCORE}} = W_1^{\text{SAF}} = \alpha$, and $R_0^{\text{SCORE}} = R_0^{\text{SAF}} = 0$. From the update rule, $\alpha_1^{\text{SCORE}} = \alpha_1^{\text{SAF}}$. Both inequalities hold.

**Inductive step:** Assume $\alpha_j^{\text{SCORE}} \geq \alpha_j^{\text{SAF}}$ and $W_j^{\text{SCORE}} \geq W_j^{\text{SAF}}$ for all $1 \leq j \leq t$.

1. *Rejections:* For any $j \leq t$, the condition $\alpha_j^{\text{SCORE}} \geq \alpha_j^{\text{SAF}}$ implies that SCORE is more likely to reject. Thus, $R_t^{\text{SCORE}} \geq R_t^{\text{SAF}}$.

2. *Wealth:* We show $W_{t+1}^{\text{SCORE}} \geq W_{t+1}^{\text{SAF}}$. From Step 2, we have $\frac{W_{t+1}^{\text{SCORE}}}{W_t^{\text{SCORE}}} \geq \frac{W_{t+1}^{\text{SAF}}}{W_t^{\text{SAF}}}$. Combining this with the inductive hypothesis $W_t^{\text{SCORE}} \geq W_t^{\text{SAF}}$, we obtain:

$$W_{t+1}^{\text{SCORE}} \geq W_t^{\text{SCORE}} \cdot \frac{W_{t+1}^{\text{SAF}}}{W_t^{\text{SAF}}} \geq W_t^{\text{SAF}} \cdot \frac{W_{t+1}^{\text{SAF}}}{W_t^{\text{SAF}}} = W_{t+1}^{\text{SAF}}.$$

Finally, for the threshold $\alpha_{t+1}$:

$$\alpha_{t+1}^{\text{SCORE}} = \omega_{t+1}(1 - \lambda_{t+1})(R_t^{\text{SCORE}} + 1)W_{t+1}^{\text{SCORE}}.$$

Since $R_t^{\text{SCORE}} \geq R_t^{\text{SAF}}$ and $W_{t+1}^{\text{SCORE}} \geq W_{t+1}^{\text{SAF}}$, it follows that $\alpha_{t+1}^{\text{SCORE}} \geq \alpha_{t+1}^{\text{SAF}}$. The induction holds for $t + 1$.

*Step 4: Strict Inequality Conditions.* We define two cases where a strict advantage is initialized at some step $j < t$:

**Case 1: Differential Rejection ($1/\alpha_j^{\text{SAF}} \geq e_j > 1/\alpha_j^{\text{SCORE}}$).** In this scenario, SCORE rejects ($\delta_j^{\text{SCORE}} = 1$) while e-SAFFRON does not ($\delta_j^{\text{SAF}} = 0$). Consequently,

$$R_j^{\text{SCORE}} > R_j^{\text{SAF}}.$$

Since we have shown in Step 3 that $\alpha_i^{\text{SCORE}} \geq \alpha_i^{\text{SAF}}$ for all steps $i$, any rejection made by e-SAFFRON is also made by SCORE. Thus, the gap in total rejections can never be reversed, so $R_{t-1}^{\text{SCORE}} > R_{t-1}^{\text{SAF}}$ holds for all future times $t > j$. Now, consider the update formula for $\alpha_t$:

$$\alpha_t = \omega_t(1 - \lambda_t)(R_{t-1} + 1)W_t.$$

Comparing the terms for SCORE and e-SAFFRON:

- The constants $\omega_t(1 - \lambda_t)$ are identical.

- The rejection term satisfies $(R_{t-1}^{\text{SCORE}} + 1) > (R_{t-1}^{\text{SAF}} + 1)$ as derived above.

- The wealth term satisfies $W_t^{\text{SCORE}} \geq W_t^{\text{SAF}}$ from Step 3.

Since all terms are non-negative and one is strictly larger, we have $\alpha_t^{\text{SCORE}} > \alpha_t^{\text{SAF}}$.

**Case 2: Strict Wealth Gain** ($0 < e_j < 1/\lambda_j$)**.** In this regime, SCORE loses strictly less wealth than e-SAFFRON. The multiplicative updates at step $j$ are:

$$W_{j+1}^{\text{SAF}} = W_j^{\text{SAF}}(1 - \omega_j), \quad W_{j+1}^{\text{SCORE}} = W_j^{\text{SCORE}}(1 - \omega_j(1 - \lambda_j e_j)).$$

Since $0 < \lambda_j e_j < 1$, we have $1 - \omega_j(1 - \lambda_j e_j) > 1 - \omega_j$. Combined with $W_j^{\text{SCORE}} \geq W_j^{\text{SAF}}$, this implies:

$$W_{j+1}^{\text{SCORE}} > W_{j+1}^{\text{SAF}}.$$

This strict gap propagates to future steps $t > j$ through the wealth recurrence. Specifically,

$$W_t = W_{j+1} \prod_{k=j+1}^{t-1} (\text{update factor}_k).$$

Since the SCORE update factors are universally greater than or equal to e-SAFFRON's (as shown in Step 2), we have:

$$W_t^{\text{SCORE}} \geq W_{j+1}^{\text{SCORE}} \prod(\dots)^{\text{SAF}} > W_{j+1}^{\text{SAF}} \prod(\dots)^{\text{SAF}} = W_t^{\text{SAF}}.$$

Finally, using the $\alpha_t$ update formula again:

$$\alpha_t^{\text{SCORE}} = \underbrace{\omega_t(1 - \lambda_t)}_{\text{same}} \underbrace{(R_{t-1}^{\text{SCORE}} + 1)}_{\geq} \underbrace{W_t^{\text{SCORE}}}_{> \text{ strictly}} W_t^{\text{SAF}}.$$

Thus, $\alpha_t^{\text{SCORE}} > \alpha_t^{\text{SAF}}$. In both cases, strict dominance is guaranteed. $\qquad\square$

### D.7. Proof of Theorem 5.2

*Proof.* We start from the definition

$$\text{FDR}(t) = \sum_{j \in \mathcal{H}_0} \mathbb{E}\left[\frac{\delta_j}{R_t \vee 1}\right].$$

Fix a null index $j \in \mathcal{H}_0$. Let $f(e_j) = \delta_j + O_j$. Observe that given $\mathcal{F}_{j-1}$, $f(e_j)$ is a non-decreasing function of $e_j$. Let $g(R_t) = -\frac{1}{R_t \vee 1}$. This is a non-decreasing function of $R_t$. By Assumption 5.1, $(e_j, R_t)$ are conditionally PQD given $\mathcal{F}_{j-1}$. By Lehmann's Lemma 1 (i) (Lehmann, 2011), $(f(e_j), g(R_t))$ are conditionally PQD given $\mathcal{F}_{j-1}$. Furthermore, by Lemma 3 in Lehmann (2011), we have $\text{Cov}(f(e_j), g(R_t) \mid \mathcal{F}_{j-1}) \geq 0$. Substituting the functions back:

$$\text{Cov}\left(\delta_j + O_j, -\frac{1}{R_t \vee 1} \,\Big|\, \mathcal{F}_{j-1}\right) \geq 0 \implies \text{Cov}\left(\delta_j + O_j, \frac{1}{R_t \vee 1} \,\Big|\, \mathcal{F}_{j-1}\right) \leq 0.$$

Using this negative covariance:

$$\mathbb{E}\left[\frac{\delta_j + O_j}{R_t \vee 1} \,\Big|\, \mathcal{F}_{j-1}\right] = \mathbb{E}[\delta_j + O_j \mid \mathcal{F}_{j-1}]\mathbb{E}\left[\frac{1}{R_t \vee 1} \mid \mathcal{F}_{j-1}\right] + \text{Cov}(\dots)$$

$$\leq \mathbb{E}[\delta_j + O_j \mid \mathcal{F}_{j-1}]\mathbb{E}\left[\frac{1}{R_t \vee 1} \mid \mathcal{F}_{j-1}\right].$$

By Lemma 3.1 (conditioned on $\mathcal{F}_{j-1}$), $\mathbb{E}[\delta_j + O_j \mid \mathcal{F}_{j-1}] \leq \mathbb{E}[\alpha_j e_j \mid \mathcal{F}_{j-1}] \leq \alpha_j$. Thus,

$$\mathbb{E}\left[\frac{\delta_j + O_j}{R_t \vee 1}\right] \leq \mathbb{E}\left[\alpha_j \cdot \mathbb{E}\left(\frac{1}{R_t \vee 1} \mid \mathcal{F}_{j-1}\right)\right] = \mathbb{E}\left[\frac{\alpha_j}{R_t \vee 1}\right].$$

Subtracting the overshoot term:

$$\mathbb{E}\left[\frac{\delta_j}{R_t \vee 1}\right] = \mathbb{E}\left[\frac{\delta_j + O_j}{R_t \vee 1}\right] - \mathbb{E}\left[\frac{O_j}{R_t \vee 1}\right] \leq \mathbb{E}\left[\frac{\alpha_j - O_j}{R_t \vee 1}\right] \leq \mathbb{E}\left[\frac{(\alpha_j - O_j)_+}{R_t \vee 1}\right].$$

Therefore, summing over all null indices and using linearity of expectation yields

$$\mathrm{FDR}(t) = \sum_{j \in \mathcal{H}_0} \mathbb{E}\left[\frac{\delta_j}{R_t \vee 1}\right] \leq \sum_{j \in \mathcal{H}_0} \mathbb{E}\left[\frac{(\alpha_j - O_j)_+}{R_t \vee 1}\right]$$

$$\leq \sum_{j=1}^{t} \mathbb{E}\left[\frac{(\alpha_j - O_j)_+}{R_t \vee 1}\right] = \mathbb{E}\left[\widehat{\mathrm{FDP}}_{\mathrm{S^+\text{-}LORD}}(t)\right].$$

$\square$

### D.8. Proof of Proposition 5.3

*Proof.* We prove $W_{t+1} \geq 0$ by induction on $t$, where $W_t = \alpha - \sum_{j=1}^{t-1} \frac{(\alpha_j^{\mathrm{S^+\text{-}LORD}} - O_j)_+}{R_{t-1} \vee 1}$.

*Base case ($t = 0$):* By definition, $W_1 = \alpha(0 \vee 1) - 0 = \alpha > 0$.

*Inductive step:* Assume $W_t \geq 0$. We show that $W_{t+1} \geq 0$. Let $S_t = W_t(R_{t-1} \vee 1)$ be the total unspent budget. The update rule $\alpha_t^{\mathrm{S^+\text{-}LORD}} = \omega_t(R_{t-1} \vee 1)W_t$ can be written as $\alpha_t^{\mathrm{S^+\text{-}LORD}} = \omega_t S_t$. From the recurrence for $W_{t+1}$:

$$W_{t+1}(R_t \vee 1) = \alpha(R_t \vee 1) - \sum_{j=1}^{t}(\alpha_j^{\mathrm{S^+\text{-}LORD}} - O_j)_+ = S_{t+1}.$$

We can express $S_{t+1}$ in terms of $S_t$:

$$S_{t+1} = S_t + \alpha\big[(R_t \vee 1) - (R_{t-1} \vee 1)\big] - (\alpha_t^{\mathrm{S^+\text{-}LORD}} - O_t)_+.$$

We now make two key observations:

1. Since $R_t \geq R_{t-1}$, we have $(R_t \vee 1) \geq (R_{t-1} \vee 1)$. Thus, $\alpha\big[(R_t \vee 1) - (R_{t-1} \vee 1)\big] \geq 0$.

2. By the definition of the update rule and the fact that $(\alpha_t^{\mathrm{S^+\text{-}LORD}} - O_t)_+ \leq \alpha_t^{\mathrm{S^+\text{-}LORD}}$, we have $(\alpha_t^{\mathrm{S^+\text{-}LORD}} - O_t)_+ \leq \alpha_t^{\mathrm{S^+\text{-}LORD}} = \omega_t S_t$.

Combining these observations, we obtain:

$$S_{t+1} \geq S_t + 0 - \omega_t S_t = S_t(1 - \omega_t).$$

By the inductive hypothesis, $W_t \geq 0$ implies $S_t \geq 0$. Since $\omega_t \in (0, 1)$, we have $S_{t+1} \geq 0$, which implies $W_{t+1} \geq 0$.

This completes the proof that the update rule ensures $\widehat{\mathrm{FDP}}_{\mathrm{S^+\text{-}LORD}}(t) \leq \alpha$ for all $t$. $\square$

### D.9. Proof of Theorem 5.4

*Proof.* We first prove the FDR control, then show that the update rule ensures the estimator is bounded by $\alpha$.

**Part 1: FDR Control.** Let $C_j^{\mathrm{S^+\text{-}SAF}} = \left(\frac{\alpha_j(1-\lambda_j e_j)}{1-\lambda_j} - O_j\right)_+$. We aim to bound $\mathbb{E}[\delta_j/(R_t \vee 1)]$ by $\mathbb{E}[C_j^{\mathrm{S^+\text{-}SAF}}/(R_t \vee 1)]$. Consider the function $h(e_j) = \delta_j - C_j^{\mathrm{S^+\text{-}SAF}}$. As shown in standard validity proofs, we have $\mathbb{E}[h(e_j) \mid \mathcal{F}_{j-1}] \leq 0$ under the null. Observe that given $\mathcal{F}_{j-1}$, $h(e_j)$ is a non-decreasing function of $e_j$. Let $g(R_t) = -\frac{1}{R_t \vee 1}$, which is non-decreasing in $R_t$. By Assumption 5.1, $(e_j, R_t)$ are conditionally PQD. Thus, $\mathrm{Cov}(h(e_j), g(R_t) \mid \mathcal{F}_{j-1}) \geq 0$. This implies $\mathrm{Cov}(h(e_j), \frac{1}{R_t \vee 1} \mid \mathcal{F}_{j-1}) \leq 0$. Then:

$$\mathbb{E}\left[\frac{\delta_j - C_j^{\mathrm{S^+\text{-}SAF}}}{R_t \vee 1} \;\middle|\; \mathcal{F}_{j-1}\right] = \mathbb{E}[h(e_j) \mid \mathcal{F}_{j-1}]\mathbb{E}\left[\frac{1}{R_t \vee 1} \;\middle|\; \mathcal{F}_{j-1}\right] + \mathrm{Cov}(\dots)$$

$$\leq 0 \cdot \mathbb{E}\left[\frac{1}{R_t \vee 1} \;\middle|\; \mathcal{F}_{j-1}\right] + 0 = 0.$$

Taking unconditional expectation and summing over all null indices gives the desired bound. Concretely,

$$0 \geq \mathbb{E}\left[\mathbb{E}\left[\frac{\delta_j - C_j^{\text{S}^+\text{-SAF}}}{R_t \vee 1} \,\bigg|\, \mathcal{F}_{j-1}\right]\right] = \mathbb{E}\left[\frac{\delta_j - C_j^{\text{S}^+\text{-SAF}}}{R_t \vee 1}\right]$$

$$\implies \mathbb{E}\left[\frac{\delta_j}{R_t \vee 1}\right] \leq \mathbb{E}\left[\frac{C_j^{\text{S}^+\text{-SAF}}}{R_t \vee 1}\right].$$

Summing over $j \in \mathcal{H}_0$ and using the nonnegativity of $C_j^{\text{S}^+\text{-SAF}}$ yields

$$\text{FDR}(t) = \sum_{j \in \mathcal{H}_0} \mathbb{E}\left[\frac{\delta_j}{R_t \vee 1}\right] \leq \sum_{j \in \mathcal{H}_0} \mathbb{E}\left[\frac{C_j^{\text{S}^+\text{-SAF}}}{R_t \vee 1}\right]$$

$$\leq \sum_{j=1}^{t} \mathbb{E}\left[\frac{C_j^{\text{S}^+\text{-SAF}}}{R_t \vee 1}\right]$$

$$= \mathbb{E}[\widehat{\text{FDP}}_{\text{S}^+\text{-SAF}}(t)].$$

**Part 2: Update Rule Validity.** We prove by induction that $W_{t+1} \geq 0$ for all $t$, where $W_t = \alpha - \sum_{j=1}^{t-1} \frac{C_j^{\text{S}^+\text{-SAF}}}{R_{t-1} \vee 1}$.

Let $X_j = \left(\frac{\alpha_j^{\text{S}^+\text{-SAF}}(1-\lambda_j e_j)}{1-\lambda_j} - O_j\right)_+$.

*Base case (t=1):* $W_1 = \alpha(0 \vee 1) - 0 = \alpha > 0$.

*Inductive step:* Assume $W_t \geq 0$. We show $W_{t+1} \geq 0$. Let $S_t = W_t(R_{t-1} \vee 1)$. Recall the definition

$$X_t = \left(\frac{\alpha_t^{\text{S}^+\text{-SAF}}(1-\lambda_t e_t)}{1-\lambda_t} - O_t\right)_+,$$

where $O_t = (\alpha_t^{\text{S}^+\text{-SAF}} e_t - 1)_+ \geq 0$. Since $(a-b)_+ \leq a_+$ for $b \geq 0$, we obtain

$$X_t \leq \left(\frac{\alpha_t^{\text{S}^+\text{-SAF}}(1-\lambda_t e_t)}{1-\lambda_t}\right)_+ \leq \frac{\alpha_t^{\text{S}^+\text{-SAF}}(1-\lambda_t e_t)}{1-\lambda_t} \leq \frac{\alpha_t^{\text{S}^+\text{-SAF}}}{1-\lambda_t}.$$

Finally, using the update rule $\alpha_t^{\text{S}^+\text{-SAF}} = \omega_t(1-\lambda_t)(R_{t-1} \vee 1)W_t$ we get

$$\frac{\alpha_t^{\text{S}^+\text{-SAF}}}{1-\lambda_t} = \omega_t(R_{t-1} \vee 1)W_t = \omega_t S_t,$$

so the claimed bound $X_t \leq \omega_t S_t$ follows. The recurrence for $S_t$ is:

$$S_{t+1} = \alpha(R_t \vee 1) - \sum_{j=1}^{t} X_j.$$

This can be rewritten in terms of $S_t$:

$$S_{t+1} = \alpha(R_t \vee 1) - \left(\sum_{j=1}^{t-1} X_j + X_t\right)$$

$$= \left(\alpha(R_{t-1} \vee 1) - \sum_{j=1}^{t-1} X_j\right) + \alpha\big[(R_t \vee 1) - (R_{t-1} \vee 1)\big] - X_t$$

$$= S_t + \alpha\big[(R_t \vee 1) - (R_{t-1} \vee 1)\big] - X_t.$$

We now make two key observations:

1. Since $R_t \geq R_{t-1}$, the function $x \mapsto x \vee 1$ is non-decreasing. Thus, $\alpha\big[(R_t \vee 1) - (R_{t-1} \vee 1)\big] \geq 0$.

2. The cost term $X_t$ is bounded by $X_t \leq \omega_t S_t$.

Combining these observations:

$$S_{t+1} \geq S_t + 0 - \omega_t S_t = S_t(1 - \omega_t).$$

By the inductive hypothesis $W_t \geq 0 \implies S_t \geq 0$ and since $\omega_t \in (0, 1)$, we have $S_{t+1} \geq 0 \implies W_{t+1} \geq 0$, completing the induction.

This completes the proof that the update rule ensures $\widehat{\mathrm{FDP}}_{\text{S}^+\text{-SAF}}(t) \leq \alpha$ for all $t$. $\qquad\square$

## D.10. Proof of Proposition A.1

*Proof.* We condition on $\mathcal{F}_{j-1}$. The remaining random variables $e_j, \ldots, e_t$ are independent. We use the following standard association result: by the FKG inequality, if $X_1, \ldots, X_n$ are independent and $f, g$ are coordinate-wise non-decreasing functions, then

$$\mathrm{Cov}(f(\mathbf{X}), g(\mathbf{X})) \geq 0$$

(Fortuin et al., 1971).

Define $\mathbf{e} = (e_j, \ldots, e_t)$. The coordinate $e_j$ is trivially non-decreasing in $\mathbf{e}$. We next show that $R_t$ is also coordinate-wise non-decreasing in $\mathbf{e}$. Let $\mathbf{e} \preceq \mathbf{e}'$. At step $j$, the threshold $\alpha_j$ is fixed after conditioning on $\mathcal{F}_{j-1}$, so

$$\mathbb{I}(e_j \geq 1/\alpha_j) \leq \mathbb{I}(e_j' \geq 1/\alpha_j).$$

For $k > j$, suppose the rejection history under $\mathbf{e}'$ dominates that under $\mathbf{e}$ up to time $k - 1$. By the assumed monotonicity of the update rule, $\alpha_k(\mathbf{e}') \geq \alpha_k(\mathbf{e})$, and hence $1/\alpha_k(\mathbf{e}') \leq 1/\alpha_k(\mathbf{e})$. Together with $e_k' \geq e_k$, any rejection under $\mathbf{e}$ at time $k$ is also a rejection under $\mathbf{e}'$. Induction over $k$ gives $R_t(\mathbf{e}') \geq R_t(\mathbf{e})$.

Therefore, for any $x, y$, the indicators $\mathbb{I}(e_j > x)$ and $\mathbb{I}(R_t > y)$ are coordinate-wise non-decreasing functions of the independent vector $\mathbf{e}$. The FKG inequality yields

$$\mathbb{P}(e_j > x, R_t > y \mid \mathcal{F}_{j-1}) \geq \mathbb{P}(e_j > x \mid \mathcal{F}_{j-1})\mathbb{P}(R_t > y \mid \mathcal{F}_{j-1}),$$

which is equivalent to PQD of $(e_j, R_t)$. Thus Assumption 5.1 holds. $\qquad\square$

## D.11. Proof of Proposition A.2

*Proof.* Since $\alpha_t = \omega_t S_t$ (for LORD) or $\alpha_t = \omega_t(1 - \lambda_t)S_t$ (for SAFFRON), and $\omega_t, \lambda_t$ are independent of the rejection history, it suffices to show that the unspent budget (denote it by) $S_t$ is non-decreasing in the rejection history $(\delta_1, \ldots, \delta_{t-1})$.

We prove by induction on $s$ that $S_s' \geq S_s$ for all $s \geq 1$, where primes denote quantities under a history $\delta'$ that dominates $\delta$ (i.e., $\delta_k' \geq \delta_k$ for all $k$). The base case $s = 1$ is trivial because $S_1 = \alpha(R_0 \vee 1) = \alpha$ regardless of history, so $S_1' = S_1$.

Assume that for some $j \geq 1$ we have $S_j' \geq S_j$ (induction hypothesis). Then $\alpha_j' \geq \alpha_j$. Write the unspent-budget update recursively:

$$S_{j+1} = S_j + \alpha\big[(R_j \vee 1) - (R_{j-1} \vee 1)\big] - D_j,$$

where $D_j$ is the cost term. For LORD: $D_j = (\alpha_j - O_j)_+$. For SAFFRON: $D_j = (\frac{\alpha_j(1-\lambda_j e_j)}{1-\lambda_j} - O_j)_+$.

Let $\Delta S_j = S_j' - S_j \geq 0$. Then

$$\Delta S_{j+1} = \Delta S_j + \alpha\Big\{\big[(R_j' \vee 1) - (R_{j-1}' \vee 1)\big] - \big[(R_j \vee 1) - (R_{j-1} \vee 1)\big]\Big\} - (D_j' - D_j). \qquad (10)$$

**Rejection term.** The term in curly braces represents the difference in effective unspent-budget increments. Let $\Delta\Psi_k = (R_k \vee 1) - (R_{k-1} \vee 1)$. We claim that $\Delta\Psi_j' \geq \Delta\Psi_j$. First, observe that $S_j' \geq S_j$ implies $\alpha_j' \geq \alpha_j$. The rejection condition is $e_j \geq 1/\alpha_j$. Since $1/\alpha_j' \leq 1/\alpha_j$, it follows that $\delta_j' \geq \delta_j$. Now consider the value of $\Delta\Psi_j$:

- If $R_{j-1} > 0$, then $(R_{j-1} \vee 1) = R_{j-1}$ and $(R_j \vee 1) = R_{j-1} + \delta_j$. Thus $\Delta\Psi_j = \delta_j$.

- If $R_{j-1} = 0$, then $(R_{j-1} \vee 1) = 1$. Since $R_j = \delta_j \in \{0, 1\}$, we have $(R_j \vee 1) = 1$. Thus $\Delta\Psi_j = 1 - 1 = 0$.

We compare $\Delta\Psi'_j$ and $\Delta\Psi_j$:

1. If $R_{j-1} > 0$, then $R'_{j-1} \geq R_{j-1} > 0$, so both increments are simply the rejection indicators: $\Delta\Psi'_j - \Delta\Psi_j = \delta'_j - \delta_j \geq 0$.

2. If $R_{j-1} = 0$, then $\Delta\Psi_j = 0$. Since $\Delta\Psi'_j$ is always non-negative (as $R'$ is non-decreasing), the difference is $\geq 0$.

Thus, the rejection term is non-negative.

**Cost difference.** We bound $D'_j - D_j$ using the Lipschitz property of the cost function $D(\alpha)$.

- For LORD, $D(\alpha) = (\alpha - (\alpha e_j - 1)_+)_+$. The slope of $D(\alpha)$ with respect to $\alpha$ is bounded above by 1 (for instance, when $e_j = 0$, $D(\alpha) = \alpha$). Thus $D(\alpha') - D(\alpha) \leq 1 \cdot (\alpha' - \alpha)$. Since $\alpha_j = \omega_j S_j$, we have $D'_j - D_j \leq \omega_j \Delta S_j$.

- For SAFFRON, $D(\alpha) = \left( \frac{\alpha(1 - \lambda_j e_j)}{1 - \lambda_j} - (\alpha e_j - 1)_+ \right)_+$. The slope is bounded above by $\frac{1}{1 - \lambda_j}$ (attained when $e_j = 0$). Thus $D(\alpha') - D(\alpha) \leq \frac{1}{1 - \lambda_j}(\alpha' - \alpha)$. Since $\alpha_j = \omega_j(1 - \lambda_j)S_j$, we have $\alpha'_j - \alpha_j = \omega_j(1 - \lambda_j)\Delta S_j$. Substituting this yields

$$D'_j - D_j \leq \frac{1}{1 - \lambda_j} \cdot \omega_j(1 - \lambda_j)\Delta S_j = \omega_j \Delta S_j.$$

In both cases, we have established $D'_j - D_j \leq \omega_j \Delta S_j$. Substituting back:

$$\Delta S_{j+1} \geq \Delta S_j + 0 - \omega_j \Delta S_j = (1 - \omega_j)\Delta S_j.$$

Since $\omega_j \in (0, 1)$, we have $1 - \omega_j > 0$. Together with the induction hypothesis $\Delta S_j \geq 0$, we obtain $\Delta S_{j+1} \geq 0$.

Thus by induction, $S'_s \geq S_s$ for all $s$. In particular, for $s = t$ we have $S'_t \geq S_t$, and consequently $\alpha'_t \geq \alpha_t$. This completes the proof. $\square$

### D.12. Proof of Theorem A.6

*Proof.* Fix a null index $j$. We apply the covariance decomposition to $\mathbb{E}[\delta_j/(R_t \vee 1)]$. Let $f(p_j) = \delta_j = \mathbb{I}(p_j \leq \alpha_j)$. This is a non-increasing function of $p_j$. Let $g(R_t) = \frac{1}{R_t \vee 1}$. This is a non-increasing function of $R_t$. By Assumption A.5, $(p_j, R_t)$ are conditionally NQD given $\mathcal{F}_{j-1}$. Since both $f$ and $g$ are non-increasing, any pair $(p_j, R_t)$ satisfying NQD will result in non-positive covariance between $f(p_j)$ and $g(R_t)$. Thus $\text{Cov}(\delta_j, \frac{1}{R_t \vee 1} \mid \mathcal{F}_{j-1}) \leq 0$.

Decomposition:

$$\mathbb{E}\left[ \frac{\delta_j}{R_t \vee 1} \mid \mathcal{F}_{j-1} \right] = \text{Cov}\left( \delta_j, \frac{1}{R_t \vee 1} \mid \mathcal{F}_{j-1} \right) + \mathbb{E}[\delta_j \mid \mathcal{F}_{j-1}] \cdot \mathbb{E}\left[ \frac{1}{R_t \vee 1} \mid \mathcal{F}_{j-1} \right]$$

$$\leq \mathbb{E}[\delta_j \mid \mathcal{F}_{j-1}] \cdot \mathbb{E}\left[ \frac{1}{R_t \vee 1} \mid \mathcal{F}_{j-1} \right]$$

$$\leq \alpha_j \cdot \mathbb{E}\left[ \frac{1}{R_t \vee 1} \mid \mathcal{F}_{j-1} \right]$$

$$= \mathbb{E}\left[ \frac{\alpha_j}{R_t \vee 1} \mid \mathcal{F}_{j-1} \right].$$

Taking unconditional expectation of the displayed inequality gives

$$\mathbb{E}\left[ \frac{\delta_j}{R_t \vee 1} \right] \leq \mathbb{E}\left[ \frac{\alpha_j}{R_t \vee 1} \right].$$

Summing over all null indices and using linearity of expectation yields

$$\text{FDR}(t) = \sum_{j \in \mathcal{H}_0} \mathbb{E}\left[ \frac{\delta_j}{R_t \vee 1} \right] \leq \sum_{j \in \mathcal{H}_0} \mathbb{E}\left[ \frac{\alpha_j}{R_t \vee 1} \right] \leq \sum_{j=1}^{t} \mathbb{E}\left[ \frac{\alpha_j}{R_t \vee 1} \right] = \mathbb{E}\left[ \frac{1}{R_t \vee 1} \sum_{j=1}^{t} \alpha_j \right].$$

$\square$

## D.13. Proof of Theorem A.7

*Proof.* Let $h(p_j) = \delta_j - C_j^{\text{SAF}} = \mathbb{I}(p_j \leq \alpha_j) - \frac{\alpha_j \mathbb{I}\{p_j > \lambda_j\}}{1 - \lambda_j}$. We verify that $h(p_j)$ is a non-increasing function of $p_j$. If $p_j \leq \alpha_j$: $h(p_j) = 1 - 0 = 1$. If $\alpha_j < p_j \leq \lambda_j$: $h(p_j) = 0 - 0 = 0$. If $p_j > \lambda_j$: $h(p_j) = 0 - \frac{\alpha_j}{1 - \lambda_j} < 0$. Thus $h(p_j)$ is non-increasing. Let $g(R_t) = \frac{1}{R_t \vee 1}$, which is non-increasing in $R_t$. By Assumption A.5 (NQD), since both functions are non-increasing, their covariance is non-positive: $\text{Cov}(h(p_j), g(R_t) \mid \mathcal{F}_{j-1}) \leq 0$. Thus,

$$\mathbb{E}\left[\frac{\delta_j - C_j^{\text{SAF}}}{R_t \vee 1} \,\Big|\, \mathcal{F}_{j-1}\right] = \text{Cov}\left(h(p_j), \frac{1}{R_t \vee 1} \,\Big|\, \mathcal{F}_{j-1}\right) + \mathbb{E}[h(p_j) \mid \mathcal{F}_{j-1}] \cdot \mathbb{E}\left[\frac{1}{R_t \vee 1} \mid \mathcal{F}_{j-1}\right]$$

$$\leq \mathbb{E}[h(p_j) \mid \mathcal{F}_{j-1}] \cdot \mathbb{E}\left[\frac{1}{R_t \vee 1} \mid \mathcal{F}_{j-1}\right].$$

Under the null with super-uniform $p$-values, conditioning on $\mathcal{F}_{j-1}$ we have

$$\mathbb{E}[h(p_j) \mid \mathcal{F}_{j-1}] = \mathbb{P}(p_j \leq \alpha_j \mid \mathcal{F}_{j-1}) - \frac{\alpha_j}{1 - \lambda_j}\mathbb{P}(p_j > \lambda_j \mid \mathcal{F}_{j-1}).$$

By super-uniformity, $\mathbb{P}(p_j \leq \alpha_j \mid \mathcal{F}_{j-1}) \leq \alpha_j$, and hence

$$\mathbb{E}[h(p_j) \mid \mathcal{F}_{j-1}] \leq \alpha_j - \frac{\alpha_j}{1 - \lambda_j}\mathbb{P}(p_j > \lambda_j \mid \mathcal{F}_{j-1}).$$

Since $\mathbb{P}(p_j > \lambda_j \mid \mathcal{F}_{j-1}) = 1 - \mathbb{P}(p_j \leq \lambda_j \mid \mathcal{F}_{j-1}) \geq 1 - \lambda_j$, we obtain

$$\mathbb{E}[h(p_j) \mid \mathcal{F}_{j-1}] \leq \alpha_j - \frac{\alpha_j}{1 - \lambda_j}(1 - \lambda_j) = 0.$$

Combining this with the covariance bound above yields the conditional inequality

$$\mathbb{E}\left[\frac{\delta_j - C_j^{\text{SAF}}}{R_t \vee 1} \,\Big|\, \mathcal{F}_{j-1}\right] \leq 0.$$

Taking unconditional expectation gives

$$\mathbb{E}\left[\frac{\delta_j}{R_t \vee 1}\right] \leq \mathbb{E}\left[\frac{C_j^{\text{SAF}}}{R_t \vee 1}\right].$$

Summing over all null indices and using linearity of expectation yields the claimed bound

$$\text{FDR}(t) = \sum_{j \in \mathcal{H}_0} \mathbb{E}\left[\frac{\delta_j}{R_t \vee 1}\right] \leq \sum_{j=1}^{t} \mathbb{E}\left[\frac{C_j^{\text{SAF}}}{R_t \vee 1}\right] = \mathbb{E}\left[\frac{1}{R_t \vee 1}\sum_{j=1}^{t} C_j^{\text{SAF}}\right].$$

$\square$

## D.14. Proof of Proposition A.8

*Proof.* Condition on $\mathcal{F}_{j-1}$. 1. $p_j$ is the first component of the remaining random vector $\mathbf{p} = (p_j, \ldots, p_t)$. Clearly $p_j$ is non-decreasing in $\mathbf{p}$. 2. We claim $R_t$ is coordinate-wise non-increasing in $\mathbf{p}$. Consider $\mathbf{p} \preceq \mathbf{p}'$. At step $j$: If $p_j \leq p_j'$, then $\delta_j(\mathbf{p}) = \mathbb{I}(p_j \leq \alpha_j) \geq \mathbb{I}(p_j' \leq \alpha_j) = \delta_j(\mathbf{p}')$. Rejection is more likely with smaller $p$. For subsequent steps $k > j$, assume $\delta_m(\mathbf{p}) \geq \delta_m(\mathbf{p}')$ for $m < k$ (more rejections in unprimed). Then $\alpha_k(\mathbf{p}) \geq \alpha_k(\mathbf{p}')$ due to monotonicity of update rule. Also $p_k \leq p_k'$. Thus $p_k \leq \alpha_k(\mathbf{p})$ is "easier" than $p_k' \leq \alpha_k(\mathbf{p}')$. Specifically, if $p_k' \leq \alpha_k(\mathbf{p}')$, then $p_k \leq p_k' \leq \alpha_k(\mathbf{p}') \leq \alpha_k(\mathbf{p})$, so $\delta_k(\mathbf{p}) = 1$. Hence $\delta_k(\mathbf{p}) \geq \delta_k(\mathbf{p}')$. This implies $R_t(\mathbf{p}) \geq R_t(\mathbf{p}')$. So $R_t$ is non-increasing in $\mathbf{p}$. Since $p_j$ is non-decreasing in $\mathbf{p}$ and $R_t$ is non-increasing in $\mathbf{p}$, and components of $\mathbf{p}$ are independent, they are NQD in the sense that $\text{Cov}(f(\mathbf{p}), g(\mathbf{p})) \leq 0$ for non-decreasing $f, g$. Applying this to indicators yields Assumption A.5. $\square$

# E. Additional Synthetic Simulation

In this section, we extend our evaluation to an Autoregressive (AR(1)) simulation setting, offering a direct comparison between the proposed SCORE algorithms and a broad range of classical $p$-value-based online FDR control methods. The AR(1) model introduces strong temporal dependence between test statistics, serving as a challenging benchmark for examining the robustness and efficiency of various procedures.

### E.1. Problem Setting

We generate a data stream $X_0, X_1, \ldots, X_T$ from an AR(1) process defined by:

$$X_t = \phi_t X_{t-1} + \epsilon_t, \quad \epsilon_t \overset{i.i.d.}{\sim} \mathcal{N}(0, 1), \tag{11}$$

for $t = 1, \ldots, T$. The process is initialized with $X_0 \sim \mathcal{N}(0, 4/3)$. Since $4/3 = 1/(1 - 0.5^2)$, this initial distribution corresponds to the stationary distribution of the AR(1) process under the null hypothesis ($\phi_t = \phi_0 = 0.5$). Consequently, under the null, the marginal distribution of $X_t$ remains stationary, i.e., $X_t \sim \mathcal{N}(0, 4/3)$ for all $t$.

At each time step $t$, we test the null hypothesis $H_{0,t} : \phi_t = \phi_0$ against the alternative $H_{1,t} : \phi_t = \phi_1$. The specific simulation parameters are:

- **AR coefficient:** Null: $\phi_0 = 0.5$, alternative: $\phi_1 = 3.0$

- **Hypothesis Generation:** The states $\theta_t \in \{0, 1\}$ are drawn i.i.d. from a Bernoulli distribution with $p_1 = 0.3$. We set $\phi_t = \phi_1$ if $\theta_t = 1$ and $\phi_t = \phi_0$ otherwise.

### E.2. Methods Compared

We evaluate a total of 12 online FDR control methods, consisting of two groups:

**1. SCORE and $e$-value Methods (6 variants).** These methods employ conditional likelihood ratio $e$-values:

$$e_t = \frac{f_{\mathcal{N}(\phi_1 X_{t-1}, 1)}(X_t)}{f_{\mathcal{N}(\phi_0 X_{t-1}, 1)}(X_t)}.$$

This group consists of the e-LORD family (e-LORD, SCORE-LORD, SCORE$^+$-LORD) and the e-SAFFRON family (e-SAFFRON, SCORE-SAFFRON, SCORE$^+$-SAFFRON).

**2. Classical $p$-value Methods (6 variants).** This group comprises 6 established $p$-value-based algorithms: LOND, LORD, SAFFRON, ADDIS, Alpha-investing, and SupLORD.

To investigate the impact of valid calibration under temporal dependence, we test these 6 classical methods using two different $p$-value constructions:

- **Conditional $p$-values:**
$$p_t^{\text{cond}} = 1 - \Phi(X_t - \phi_0 X_{t-1}).$$

  These $p$-values are super-uniform conditional on the filtration $\mathcal{F}_{t-1}$. As discussed in Section A, under the Conditional Negative Quadrant Dependence (CNQD, Assumption A.5), methods using these $p$-values are theoretically guaranteed to control FDR. This setting validates our theoretical claim that CNQD is a sufficient and strictly weaker condition than independence for valid online FDR control.

- **Marginal $p$-values:**
$$p_t^{\text{marg}} = 1 - \Phi\left(\frac{X_t}{\sqrt{4/3}}\right).$$

  Due to the stationary initialization, these $p$-values are valid marginally. However, they generally fail to satisfy the conditional super-uniformity requirement. We include these to demonstrate that marginal validity alone is insufficient for FDR control in dependent data streams.

### E.3. Results

Figures 10, 11, and 12 summarize the performance of the evaluated methods. Figure 10 presents the final power and FDR achieved at time $T$, while Figure 11 tracks the evolution of these metrics over time. Taken together, these results highlight three key findings:

1. **Validity under Dependence:** As shown in Figures 10 and 11, all 12 methods successfully control FDR when using conditionally valid statistics ($e$-values or conditional $p$-values). Figure 11 further demonstrates that FDR remains consistently below the target level $\alpha = 0.05$ throughout the entire time horizon. This validates our theoretical discussion in Section A that Conditional Negative Quadrant Dependence (CNQD) is sufficient for online FDR control, even in the presence of strong temporal correlations.

2. **High Efficiency of $e$-values:** Contrary to offline settings where $e$-value methods (e.g., e-BH(Wang & Ramdas, 2022)) often suffer significant power loss compared to $p$-value methods (e.g., BH(Benjamini & Hochberg, 1995)), our online $e$-value algorithms (SCORE/SCORE$^+$) demonstrate power comparable to, and in some cases exceeding, the best-performing classical $p$-value methods calibrated with conditional $p$-values. As evident from both Figures 10 and 11, the SCORE family consistently achieves higher power than the classical $p$-value methods throughout the sequential testing process. This suggests that in the online regime, the robustness of $e$-values does not necessarily come at the cost of statistical power.

3. **Insufficiency of Marginal Validity:** Figure 12 provides critical empirical evidence that marginal validity is inadequate for dependent data streams. Although the marginal $p$-values are uniform under the null, the lack of conditional validity leads to a catastrophic breakdown in FDR control. This aligns with the derivations in Section A, emphasizing that conditional super-uniformity is a crucial condition for valid online inference under dependence.

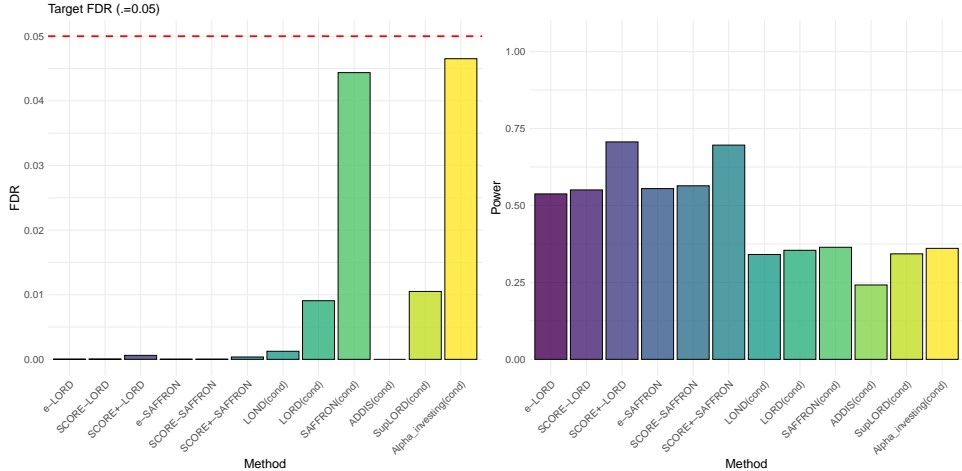

*Figure 10.* Power and FDR comparison of 12 methods under the AR(1) setting: 6 $e$-value-based methods and 6 classical $p$-value methods using conditional $p$-values. The dashed red line represents the target FDR level $\alpha = 0.05$. All methods successfully control FDR.

## F. Additional Real Data Application

We evaluate the algorithms on a real-world clinical dataset from (Golovenkin et al., 2020). The goal is to detect patients with lethal outcomes based on clinical features.

We employ the conformal $e$-value approach (Balinsky & Balinsky, 2024) to construct valid $e$-values. Specifically, we split the data into training, calibration, and test sets. We train a logistic model on the training set to output a probability score $s(x) = \hat{P}(Y = 1|x)$. For each test hypothesis $H_{0,t}$, we construct a conformal $e$-value as:

$$e_t = \frac{s(X_t)}{\frac{1}{|\mathcal{D}_{\text{cal}}|+1}\left(\sum_{Z \in \mathcal{D}_{\text{cal}}} s(Z) + s(X_t)\right)},$$

where $\mathcal{D}_{\text{cal}}$ consists of calibration samples where the null hypothesis is true (patients who survived). We set the target FDR level to $\alpha = 0.3$.

Because conformal $e$-values share a calibration sample, the independence condition used in Proposition A.1 does not automatically apply. For this reason, this application focuses on the non-retroactive SCORE variants, whose validity only requires conditional $e$-validity and does not rely on CPQD. Extending the SCORE$^+$ idea to conformal selection with shared

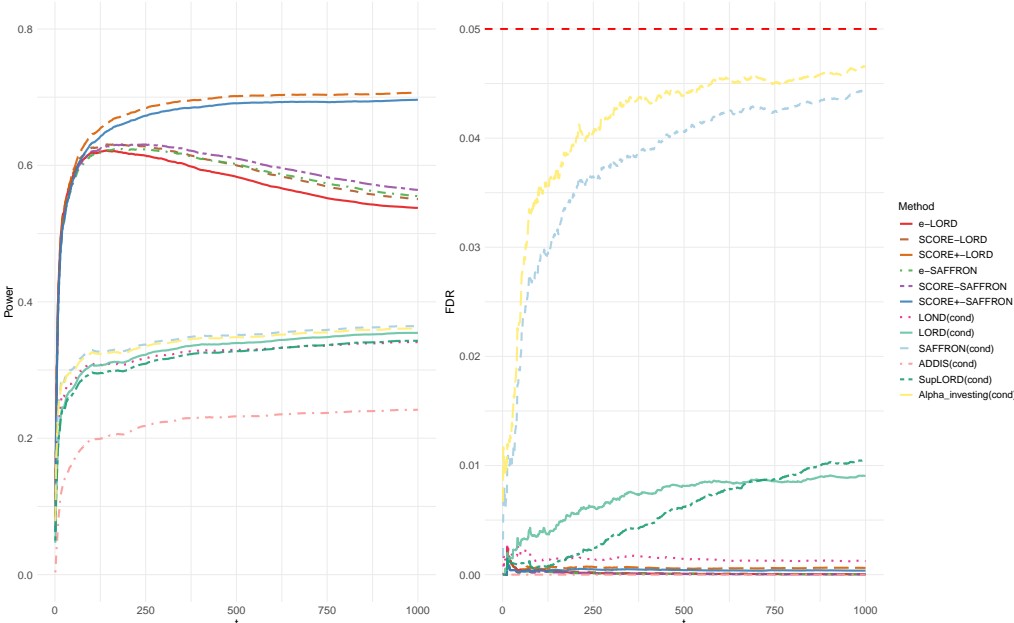

*Figure 11.* Evolution of Power and FDR over time for 12 methods under the AR(1) setting: 6 *e*-value-based methods and 6 classical *p*-value methods using *p*-values. The dashed red line represents the target FDR level $\alpha = 0.05$.

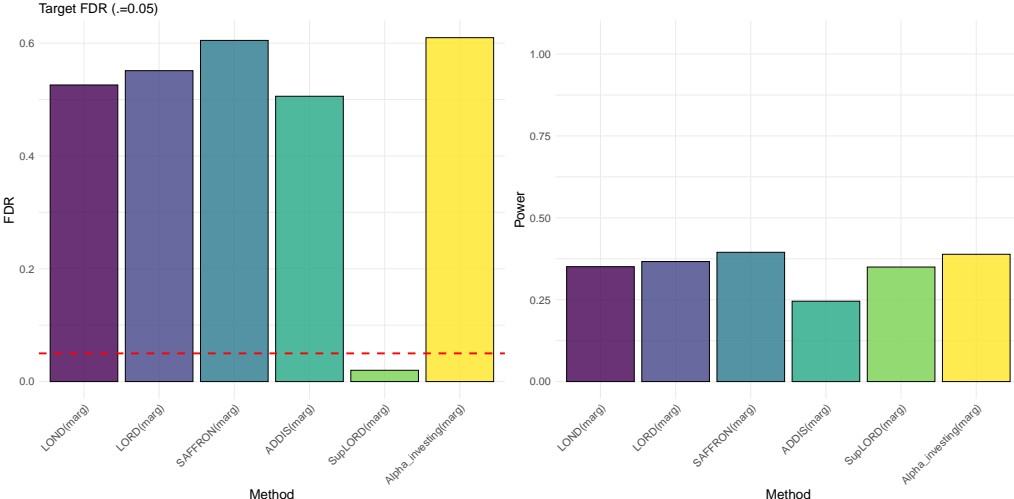

*Figure 12.* Power and FDR comparison of the same 6 classical *p*-value methods when using marginal *p*-values under the AR(1) setting. The dashed red line represents the target FDR level $\alpha = 0.05$. Note that all methods fail to control FDR, demonstrating that marginal validity is insufficient under temporal dependence.

calibration data is an interesting direction, but it requires additional dependence analysis beyond the scope of this empirical illustration.

We compare six algorithms: e-LOND, e-LORD, e-SAFFRON, and their SCORE counterparts (SCORE-LOND, SCORE-LORD, SCORE-SAFFRON).

The results, averaged over 500 random splits, are shown in Figure 13. As expected, all methods control the online FDR below the nominal level. In terms of power, SCORE-based methods outperform standard baselines. SCORE-SAFFRON achieves the highest power, which stems from its combined use of the overshoot refund mechanism and adaptive estimation of the null proportion via $\lambda$. This confirms the practical benefit of the theoretical "overshoot refund" design.

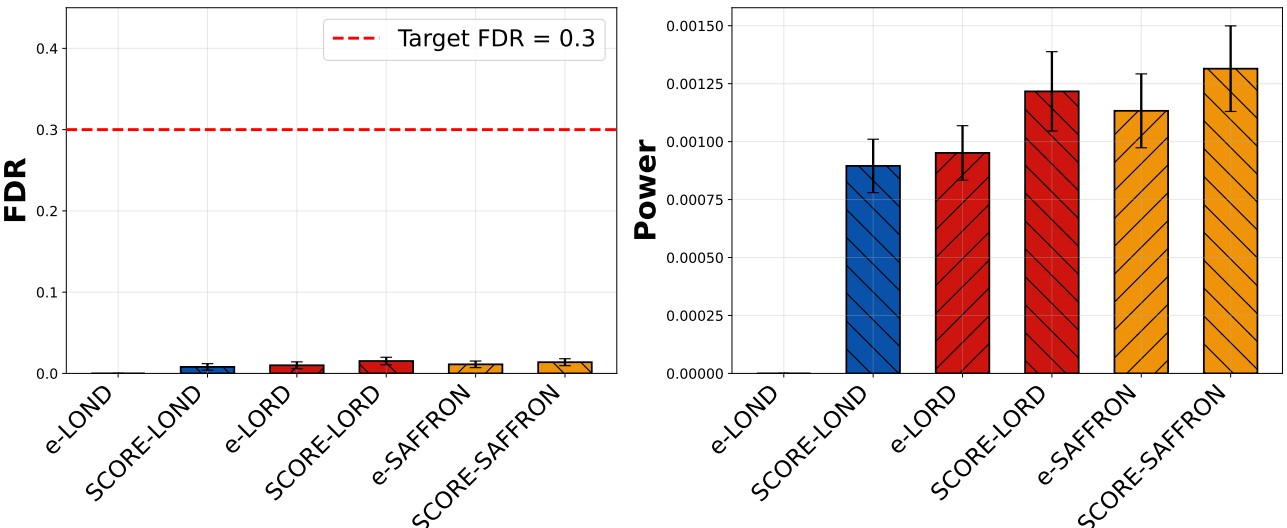

*Figure 13.* Real Data Analysis: FDR (left) and Power (right) on the Myocardial Infarction dataset. SCORE-based methods consistently achieve higher power than their standard counterparts while maintaining valid FDR control.

