# OpenReview forum: "SCORE: A Unified Framework for Overshoot Refund in Online FDR Control"
_ICML.cc/2026/Conference — ICML 2026 regular_

### Official Review · Reviewer_9dVP · 2026-03-13

**Soundness:** 4
**Presentation:** 3
**Significance:** 4
**Originality:** 4
**Overall Recommendation:** 5
**Confidence:** 4

**Summary:**

The paper considers (sequential) online multiple testing with FDR control with e-values.
They identify that existing online FDR control procedures generally rely on a very crass inequality $\mathbf{1}[e_j \geq \alpha_j^{-1}] \leq \alpha_j e_j$, which could be tightened to $\mathbf{1}[e_j \geq \alpha_j^{-1}] \leq \alpha_j e_j - (\alpha_j e_j - 1)_+$.
This has a neat interpretation on the game-theoric interpretation of e-values & multiple testing in that this tightened inequality corresponds to giving "refunds" to the alpha-wealth when the e-values have significantly more evidence against rejection than the current threshold $\alpha_j$.
The authors show that this immediately leads to strict improvements in the most prominent online FDR control procedures for e-values.
Finally, they show that under some additional structural assumptions (reminiscent of those used in FDR control for p-values) one can carry the refunds idea further, leading to even greater improvements in power.

**Compliance With Llm Reviewing Policy:**

Affirmed.

**Final Justification:**

My review was already positive, and the author's response was satisfactory. I keep my positive score.

**Key Questions For Authors:**

I'm curious about how fundamental the decreasing power over time is when not using retroactive wealth updates.

**Limitations:**

All relevant limitations were discussed appropriately.

**Strengths And Weaknesses:**

I quite liked the paper.
It makes a really good contribution: the idea of "refunds" is great (both conceptually and in terms of significance), the theory is clean, and the results (both theoretical and empirical) are good.
I think this is a significant step in the online FDR control literature.

I have only two pieces of (minor) criticism/questions.
- I found Assumption 5.1 a bit hard to grasp.
    - I think something that would have helped *a lot* would have been to emphasize that there are two indices, $j$ and $t$; since the $j$ and $t$ look visually fairly similar and are in a small font, it's easy to confuse them as both being $j$ and be stumped for a while trying to figure out what in the world the assumption is doing.
    - I also think that in the proof of Proposition A.1 (which is very instructive as it helps the reader familiarize themselves with the role of the assumption), it would be best to swap the two paragraphs so that the reader understands why talk about all the monotonicity before doing all the hard work there. I.e., first talk about the FKG inequality, then do all the hard work. Using some display equations would also be helpful.
    - I think it would also be nice to have in Appendix A some example of something that does *not* satisfy the assumption. Shouldn't be too hard, and it would definitely help the reader familiarize themselves with the assumption.
- In the numerical experiments we can see that all the methods that do not leverage retroactive updates under Assumption 5.1 (still) have decreasing power over time. Is this fundamental? Can the idea of overshoot refunds not help with this in some way? I understand, of course, that retroactive updates solve this, but they require additional structural assumptions, and intuitively I'm inclined to believe that some sort of non-retroactive refund should be able to help here.

---

> ### Author Rebuttal · Authors · 2026-03-27
>
> We sincerely thank you for your highly positive evaluation and for recognizing our work as a "significant step" in the field. Your constructive suggestions regarding the presentation and theoretical intuition are extremely valuable to us. Please find our point-by-point responses below:
>
> **1. Regarding the visual similarity of the indices:**
>
> We completely agree that the visual similarity of the indices in a small font can be confusing. To prevent this visual ambiguity and improve readability, we will update our notation in the revised manuscript by replacing $\mathcal{H}_0^t$ with $\mathcal{H}_0(t)$. This ensures that expressions like $j \in \mathcal{H}_0(t)$ are much easier to read than the original $j \in \mathcal{H}_0^t$. Furthermore, as you suggested, we will add a sentence explicitly emphasizing the distinct roles of the two indices when introducing this assumption.
>
> **2. Regarding the structure of the proof:**
>
> Swapping the paragraphs to introduce the FKG inequality first is an excellent suggestion that will indeed make the proof much easier to understand. We will restructure the proof in Proposition A.1 and add the necessary display equations exactly as recommended.
>
> **3. Regarding the CPQD Assumption:**
>
> We agree that an explicit counterexample would greatly help readers familiarize themselves with the assumption. In our **Response to Reviewer xnnP (Point 1)**, we have provided a detailed discussion and constructed a setting that violates the CPQD assumption. We kindly refer you to that section, and we will include this example in Appendix of the revised manuscript.
>
> **4. Regarding the decreasing power over time:**
>
> To directly answer your question: Yes, as you astutely observed in our plots, the decreasing power over time is indeed a fundamental challenge in online multiple testing. And yes, the idea of overshoot refunds definitely helps mitigate it.
>
> This decreasing power phenomenon is formally known as **alpha-death**. When an algorithm encounters a long sequence of nulls or weak signals without making rejections, the available $\alpha$-wealth strictly depletes. Consequently, the testing thresholds decay toward zero.
>
> How SCORE mitigates this: While SCORE cannot theoretically break this mathematical limit under the standard FDR framework, it acts as a powerful buffer. By reclaiming the "overshoot" from strong signals, SCORE consistently injects additional wealth back into the procedure. This extra wealth keeps the thresholds higher for longer, facilitating more rejections. These additional rejections structurally increase the denominator of the estimated FDP, which in turn frees up even more wealth. Therefore, SCORE can slow down the depletion process and delay the occurrence of alpha-death.
>
> An alternative type I error metric (mem-FDR): One strategy to resolve alpha-death over an infinite horizon is to shift the target from standard FDR to decaying memory FDR (mem-FDR) [1]. By introducing a discount factor $\delta \in (0, 1]$, mem-FDR assigns exponentially less weight to past discoveries and errors, allowing the available wealth to recover over time. The mem-FDR at time $t$ is formally defined as:
>
> $$
> \text{mem-FDR}(t) = \mathbb{E}\left[ \frac{\sum\_{j \in \mathcal{H}_0(t)} \delta^{t-j} \mathbf{1}(H\_j \text{ is rejected})}{\sum\_{j=1}^t \delta^{t-j} \mathbf{1}(H\_j \text{ is rejected})} \right]
> $$
>
> where $\mathcal{H}_0(t)$ is the set of true nulls up to time $t$. We adopt the standard convention that $0/0 = 0$.
>
> Because our SCORE framework acts as a highly versatile plug-in for FDP estimators, the overshoot idea can naturally be integrated into this mem-FDR framework to achieve both tighter bounds and immunity to alpha-death. We will add a detailed discussion clarifying this fundamental issue and outlining the SCORE-mem-FDR extension in the revised manuscript.
>
> **References:**
>
> [1] Ramdas, Aaditya, Fanny Yang, Martin J. Wainwright, and Michael I. Jordan. "Online control of the false discovery rate with decaying memory." *Advances in Neural Information Processing Systems*  (2017).

---

> > ### Author Rebuttal · Reviewer_9dVP · 2026-04-02
> >
> > I'd like to thank the authors for their response. The notation change for the t and j indices sounds excellent. I will keep my positive score.

---

> > > ### Author Response · Authors · 2026-04-07
> > >
> > > Thank you for your insightful comments and for your engagement in reviewing our manuscript. We will incorporate these discussions into the revision to enrich the content of our paper. Thank you once again for your support of our work!

---

### Official Review · Reviewer_82EN · 2026-03-13

**Soundness:** 3
**Presentation:** 3
**Significance:** 3
**Originality:** 3
**Overall Recommendation:** 4
**Confidence:** 2

**Summary:**

This paper proposes SCORE, a unified framework for improving e-value based online FDR control procedures. The key idea is to recover the overshoot evidence when an e-value exceeds the rejection threshold, instead of discarding it as in existing algorithms. The framework modifies the cost in the FDP estimator and can be applied as a plug-in improvement to existing methods such as e-LOND, e-LORD, and e-SAFFRON. The paper also introduces SCORE+ variants with retroactive wealth updates under mild dependence assumptions. Both theory and experiments suggest improved power while maintaining valid FDR control.

**Compliance With Llm Reviewing Policy:**

Affirmed.

**Key Questions For Authors:**

1. In the abstract, the inequality $\mathbb{I}(y \ge 1) \le y - (y-1)_+$ seems to hold only for $y \ge 0$.
2. How sensitive is the proposed framework to the quality of the e-values used in practice? For example, if the e-values are conservative, will the overshoot refund still provide meaningful gains?
3. Does the CPQD dependence assumption hold when the method is used with conformal e-values?

**Limitations:**

yes

**Strengths And Weaknesses:**

### Strengths
- The overshoot refund idea is simple and intuitive. It identifies inefficiency in existing e-value based online FDR procedures and proposes a way to recycle the excess evidence.
- The framework is general and can be applied to several existing algorithms (e-LOND, e-LORD, e-SAFFRON). This plug-in style improvement makes the idea easy to adopt.
- The paper provides clear theoretical guarantees, including finite-sample FDR control and results showing the proposed procedures dominate their original counterparts.
- In the appendix, the paper also combines the framework with conformal e-values, which is interesting, especially for applications such as online conformal selection.

### Weaknesses
- The SCORE+ variant relies on the CPQD dependence assumption. While some intuition is provided, it would be helpful to further discuss when this assumption is reasonable in practical online testing settings.

---

> ### Author Rebuttal · Authors · 2026-03-28
>
> We sincerely thank you for your encouraging evaluation, and are glad you appreciate the intuitive overshoot refund and our framework's general applicability. Please find our responses to your questions below:
>
> **1. Regarding the inequality condition in the abstract:**
>
> Thank you for catching this omission. We omitted this condition in the abstract for brevity, but the fully rigorous statement is provided in Lemma 3.1. We will update the abstract to ensure mathematical precision.
>
> **2. Regarding the sensitivity to the quality of e-values and conservative scenarios:**
>
> We completely agree with your intuition: the performance of the proposed framework is directly related to the quality of the e-values.
>
> If all e-values are conservative, there will indeed be no additional power gain (though our procedure safely reduces to the baseline without any penalty). However, such a uniformly conservative situation is rare in practice. The much more common case involves a mixture of signal strengths, where some alternative hypotheses produce strong e-values, while others hover just below the rejection threshold. In these mixed cases, our framework proves useful by recycling the excess evidence from strong signals to help detect the borderline ones.
>
> **3. Regarding when the CPQD assumption is reasonable in practical online testing settings:**
>
> In the Proposition A.1 we proves CPQD holds if e-values are independent and significance levels are non-decreasing in past decisions. In addition to the formal proof, CPQD aligns with the natural practical setting: a large $e_j$ not only directly triggers a rejection but also generates an overshoot refund that increases alpha wealth, thereby facilitating further discoveries (i.e., a large $R_t$).
>
> Violating this assumption requires a extreme scenario where a large current e-value suppresses future discoveries, which is rare in practice. Furthermore, SCORE$^+$ demonstrates remarkable empirical robustness. During our attempts to construct a counterexample, even introducing strong negative correlations between adjacent e-values failed to break the FDR control of SCORE$^+$. In fact, forcing a violation requires a highly contrived setup (as detailed in our response to **Reviewer xnnP, Point 1**).
>
> **4. Regarding whether the CPQD assumption holds with conformal e-values:**
>
> As discussed above, the CPQD assumption is generally mild. However, formally proving that the CPQD assumption holds for conformal e-values is theoretically challenging when directly applying our SCORE framework. In fact, establishing even conditional validity in this context is difficult.
>
> The core of this difficulty is that using conformal e-values (or p-values) for online testing relies on a shared calibration set (whether fixed or online updated). This shared data introduces complex dependency among the test statistics across different time.
>
> However, for your highlighted application of online conformal selection, we can adapt our SCORE framework to bypass this issue by incorporating feedback. A recent paper [1] explores this setting by assuming that at time $t$, we receive feedback revealing whether past hypotheses ($1$ to $t-1$) were null or alternative.  However, the method proposed in that paper is theoretically limited to controlling the marginal FDR (mFDR) instead of FDR.
>
> To properly address this issue and achieve control over the standard FDR, we integrate the SCORE framework into this setup and construct a feedback-enhanced FDP estimator. Let $\theta_j \in \\{0, 1\\}$ denote the true state, where $\theta_j = 0$ represents a true null. We define our estimator as:
>
> $$
> \widehat{\text{FDP}}(t) = \sum_{j=1}^{t} \frac{ (1-\theta_j)(\alpha_j-(\alpha_j e_j-1)_+)}{\tilde{R}_t \vee 1}
> $$
>
> where $\tilde{R}\_t = \sum_{j=1}^{t} \delta_j (1-\theta_j)$ represents the total number of false rejections up to time $t$.
>
> Crucially, building upon the insights of Theorem 8.2 in [2], we can prove that under this feedback mechanism, the ranks of the null score functions are mutually independent. We can utilize this property to construct rank-based conformal e-values, which directly resolves the complex dependency issue and guarantees the CPQD assumption. By combining these rank-based e-values with our feedback-enhanced FDP estimator, we can construct a rigorous online FDR control procedure.
>  We would be happy to add a dedicated discussion and an outline of this feedback-enhanced SCORE extension for conformal selection to the revised appendix, as we believe it significantly strengthens the manuscript.
>
> **Reference:**
>
> [1] Lu, Lin, Yuyang Huo, Haojie Ren, Zhaojun Wang, and Changliang Zou. "Feedback-enhanced online multiple testing with applications to conformal selection." *arXiv:2509.03297* (2025).
>
> [2] Angelopoulos, Anastasios N., Rina Foygel Barber, and Stephen Bates. "Theoretical foundations of conformal prediction." *arXiv:2411.11824* (2024).

---

> > ### Author Rebuttal · Reviewer_82EN · 2026-04-03
> >
> > Thank you for the detailed response. These changes address my questions and concerns.

---

> > > ### Author Response · Authors · 2026-04-07
> > >
> > > We sincerely appreciate your helpful comments and strong support for our work! Your feedback has helped us improve the paper, and we will incorporate these clarifications into the revision. Thank you once again for your valuable suggestions and your engagement in reviewing our work.

---

### Official Review · Reviewer_xnnP · 2026-03-13

**Soundness:** 4
**Presentation:** 3
**Significance:** 3
**Originality:** 3
**Overall Recommendation:** 5
**Confidence:** 3

**Summary:**

The paper investigates online FDR control using e-values. To enhance statistical power, the authors propose the SCORE framework. This framework leverages a sharpened mathematical bound to reclaim excess evidence  that exceeds the rejection threshold, refunding this "wasted" evidence back to the algorithm's error budget. Building on this, the authors also introduce SCORE+ variants that allow for retroactive wealth updates under a CPQD assumption. Experimental results on both synthetic and real-world datasets are conducted to demonstrate the effectiveness of the proposed methods.

**Compliance With Llm Reviewing Policy:**

Affirmed.

**Final Justification:**

The authors have successfully addressed my concerns. I recommend the paper for acceptance.

**Key Questions For Authors:**

N/A

**Limitations:**

yes

**Strengths And Weaknesses:**

Pros:

1. The objective is clear and theoretically sound. By identifying and mathematically formalizing the overshoot phenomenon, the paper provides a rigorous way to improve statistical power while maintaining valid finite-sample FDR control.
2. The proposed framework is highly versatile. It acts as a universal plug-in module that uniformly improves existing state-of-the-art e-value algorithms, such as e-LOND, e-LORD, and e-SAFFRON, without requiring fundamental structural changes to the testing procedures.


Cons:

1. The paper could benefit from an empirical robustness analysis of Assumption 5.1. While the CPQD assumption is demonstrated to be weaker than full independence, it is unclear how the SCORE+ procedures behave or degrade if this assumption is mildly violated in highly complex dependent data streams.
2. The paper lacks sufficient experimental validation across different target FDR ($\alpha$) levels. While some variations are shown, a more robust ablation study demonstrating the framework's stability across a wider range of target alpha values would strengthen the empirical claims.
3. Figures 1 and 2 are visually cluttered, with many curves plotted together. The presentation would benefit from explicitly highlighting the performance gains achieved by the proposed methods relative to their respective baselines

---

> ### Author Rebuttal · Authors · 2026-03-27
>
> We sincerely thank you for your time, careful reading, and constructive suggestions. Your insights are highly valuable and will significantly improve the quality of our paper. Please find our point-by-point responses below:
>
> **1. Regarding the analysis of Assumption 5.1 (CPQD):**
>
> Intuitively, the CPQD assumption requires a positive correlation between the current e-value $e_j$ and the total rejections $R_t$. In the context of online $\alpha$-investing, this is a highly natural and reasonable assumption. A larger observed $e_j$ not only directly increases the probability of a rejection at step $j$, immediately incrementing $R_t$, but also generates a overshoot refund. This accumulates more $\alpha$-wealth for future testing, thereby increasing the chances of rejecting subsequent hypotheses and yielding a larger $R_t$.
>
> To analyze how SCORE$^+$ degrades when this is violated, we constructed a setting specifically designed to force a negative correlation.
>
> * **Setup:** Hypotheses alternate between Nulls at odd steps and Alternatives at even steps. The e-value is defined as $e_t = \frac{1}{\alpha_t} \mathbf{1}(X_t \le \alpha_t)$. Under the null, $X_t \sim \text{Uniform}(0,1)$. Under the alternative, if any false rejection occurred before, $X_t = (1+\alpha_t)/2 > \alpha_t$. If no prior false rejection occurred, the signal remains strong, i.e., $X_t = \alpha_t/2 \le \alpha_t$.
>
> * **CPQD Violation:** For any null hypothesis $j \in \mathcal{H}_0^t$:
>   * **Case 1 ($e_j$ is small):** With probability $1-\alpha_j$, $X_j > \alpha_j$, yielding $e_j = 0$. Because no false rejection occurs, the subsequent alternatives remain strong and can be rejected. Consequently, $R_t$ tends to be **large**.
>   * **Case 2 ($e_j$ is large):** With probability $\alpha_j$, $X_j \le \alpha_j$, yielding $e_j = 1/\alpha_j$. This false rejection masks all subsequent alternatives. Consequently, $R_t$ tends to be **small**.
>
> This forces a negative correlation: a large $e_j$ leads to a small $R_t$, while a small $e_j$ leads to a large $R_t$, which directly violates the CPQD assumption.
>
> * **Empirical Conclusion:** Our simulations of this setup demonstrate that SCORE$^+$ fails to control FDR but SCORE does.
>
>  **FDR Results at T=1000 over 1000 trials ($\alpha$=0.05)**
> |Method|Mean FDR|SE of FDR|
> |-|-|-|
> | SCORE-SAFFRON | 0.0267 | 0.0044 |
> | SCORE$^+$-SAFFRON | 0.0750 | 0.0044 |
> | SCORE-LORD | 0.0347 | 0.0055 |
> | SCORE$^+$-LORD | 0.0993 | 0.0056 |
>
> It is important to emphasize that the setup constructed above is a highly contrived scenario. We test numerous other settings but all fail to break the FDR control. While this simulation delineates the theoretical boundaries of SCORE$^+$, it also highlights that the CPQD assumption is mild, and the SCORE$^+$ procedure remains robust in general settings.
>
> **2. Regarding the experimental validation across different target FDR levels:**
>
> We agree that a more robust ablation study would strengthen the empirical claims. Due to the character constraints of this rebuttal, we currently provide a representative subset of the results here, specifically focusing on the FDR and power at $T = 1000$ for target levels $\alpha \in \\{0.05,0.1, 0.2\\}$. We will expand these results with a full ablation study across a wider range of $\alpha$ values in the revised manuscript.
>
> |$\alpha$|Method|$\pi_1$=0.3 FDR|$\pi_1$=0.3 Power|$\pi_1$=0.8 FDR|$\pi_1$=0.8 Power|
> |-|-|-|-|-|-|
> |0.05|e-SAFFRON|0.0001|0.0995|0.0001|0.1984|
> |0.05|SCORE-SAFFRON|0.0002|0.1228|0.0001|0.2485|
> |0.05|SCORE$^+$-SAFFRON|0.0013|0.406|0.0007|0.4983|
> |0.05|e-LORD|0.0002|0.082|0.0001|0.0913|
> |0.05|SCORE-LORD|0.0003|0.0895|0.0001|0.1262|
> |0.05|SCORE$^+$-LORD|0.002|0.4256|0.0008|0.5095|
> |0.1|e-SAFFRON|0.0004|0.1076|0.0001|0.2176|
> |0.1|SCORE-SAFFRON|0.0004|0.1326|0.0001|0.2732|
> |0.1|SCORE$^+$-SAFFRON|0.0027|0.4459|0.0014|0.5459|
> |0.1|e-LORD|0.0005|0.0886|0.0001|0.1015|
> |0.1|SCORE-LORD|0.0006|0.0975|0.0001|0.1447|
> |0.1|SCORE$^+$-LORD|0.004|0.4678|0.0018|0.5626|
> |0.2|e-SAFFRON|0.0003|0.1201|0.0003|0.2374|
> |0.2|SCORE-SAFFRON|0.0003|0.1473|0.0003|0.2973|
> |0.2|SCORE$^+$-SAFFRON|0.0055|0.4884|0.0032|0.5912|
> |0.2|e-LORD|0.0006|0.0986|0.0004|0.1105|
> |0.2|SCORE-LORD|0.0007|0.1094|0.0005|0.1643|
> |0.2|SCORE$^+$-LORD|0.0085|0.5128|0.0046|0.6153|
>
> **3. Regarding the visual presentation of Figures 1 and 2:**
>
> We share your concern regarding the readability of these figures. Due to strict space constraints in the initial submission, we had to combine many curves into single plots, which unfortunately resulted in visual clutter. To ensure the performance gains of our methods are explicitly highlighted, we will redesign these visualizations in the revision. Specifically, we will create separate, larger figures for each class of methods, dedicating individual sub-figures to 'power vs. time' and 'power ratios'. We will ensure ample space is dedicated to this much clearer layout. We once again appreciate your constructive feedback.

---

> > ### Author Rebuttal · Reviewer_xnnP · 2026-04-01
> >
> > Thank you for the detailed response. My concerns have been adequately addressed.

---

> > > ### Author Response · Authors · 2026-04-07
> > >
> > > We sincerely thank you for taking the time to review our response and for confirming that all your concerns have been fully addressed. We deeply appreciate your constructive feedback and support for our work.

---

### Official Review · Reviewer_31Fr · 2026-03-13

**Soundness:** 3
**Presentation:** 2
**Significance:** 3
**Originality:** 3
**Overall Recommendation:** 4
**Confidence:** 3

**Summary:**

This paper proposes the SCORE framework to improve the statistical power of online multiple testing procedures based on e-values. The key idea is to replace the conventional Markov-like inequality used in existing e-value methods with a tighter bound, which enables an overshoot compensation mechanism. Specifically, when the observed e-value provides strong evidence and significantly exceeds the rejection threshold, the proposed framework reduces the accumulated testing cost by refunding the excess evidence; when the evidence lies just across the threshold, the procedure becomes equivalent to the original approach. Through this adaptive evidence compensation, the algorithm can preserve more testing budget for future hypotheses, thereby improving testing power while still maintaining valid FDR control. The paper also presents three concrete realizations of the framework—SCORE-LOND, SCORE-LORD, and SCORE-SAFFRON—demonstrating how the proposed idea can be incorporated into several existing online e-value testing procedures.

**Compliance With Llm Reviewing Policy:**

Affirmed.

**Key Questions For Authors:**

1. Are e-LORD and e-SAFFRON state-of-the-art methods for online multiple hypothesis testing? The paper mainly focuses on improving the e-LORD and e-SAFFRON procedures by introducing an adjusted-wealth mechanism. While the technical contribution is interesting and well-motivated, it is somewhat unclear how the proposed work advances the current state-of-the-art approaches in the field.

2. The proposed cross-threshold adjustment seems to provide benefits primarily when the observed e-values contain strong evidence (i.e., when the alternative hypotheses are relatively easy to detect). However, in such cases, conventional approaches may already perform well, so the advantage of the proposed method may be limited. On the other hand, when the alternative hypotheses are more challenging to detect and the evidence is close to the rejection threshold, does the proposed method effectively reduce to the conventional testing procedure without the adjusted wealth mechanism? If so, could the authors clarify in what regimes the proposed method provides the most practical benefits?

**Limitations:**

It seems there's no discussion on the limitation.

**Strengths And Weaknesses:**

### Soundness
The technical approach appears sound. The main idea is to replace the conventional Markov-like inequality used in e-value based online testing with a tighter bound, which naturally leads to an overshoot compensation mechanism. This modification is mathematically well-motivated: when the e-value significantly exceeds the rejection threshold, the framework reduces the accumulated cost spent for that test, while when the evidence lies just across the threshold the method becomes equivalent to the original approach. This design preserves the validity of the FDR control while allowing more efficient use of the testing budget.

### Presentation
The paper is generally clear, but the presentation could be improved by providing more background on e-value based hypothesis testing, particularly for readers who are more familiar with p-value based online testing procedures. In addition, a clearer explanation of the wealth (or cost-spent) accumulation mechanism used in e-value frameworks would help readers better understand how the proposed overshoot compensation affects the testing procedure.

### Significance
Multiple hypothesis testing is a well-known and important problem in statistics and machine learning, especially in modern settings involving sequential or large-scale experiments. Improving testing power while maintaining valid FDR control is a central challenge in this area. The proposed framework directly addresses this challenge and therefore has clear potential significance.

### Originality
To the best of my knowledge, the work is original in its idea of adaptively compensating the testing cost using the excess evidence contained in large e-values. This overshoot compensation mechanism provides a novel perspective on improving the efficiency of e-value based online testing procedures while maintaining theoretical guarantees.

---

> ### Author Rebuttal · Authors · 2026-03-27
>
> We sincerely thank you for your time, careful reading, and positive evaluation of our technical soundness and originality. Your insightful questions perfectly capture the core mechanisms of our framework. Here are our repsonses on the three points you mentioned:
>
> **1. Regarding the presentation:**
>
> We completely agree with your suggestion. In the revised manuscript, we will add a dedicated subsection in the background to explicitly contrast e-value and p-value based online testing and provide a clearer explanation of the wealth accumulation mechanism so that readers can better appreciate how the overshoot interacts with the testing procedure.
>
> **2. Regarding the state-of-the-art status and our broader contribution:**
>
> * **SOTA Baselines:** The field of e-value-based online FDR control is very recent. To the best of our knowledge, the theoretical foundation for this field is primarily established by two recent papers [1] and [2]. Since e-LOND, e-LORD, and e-SAFFRON are the primary methods proposed in these works, we believe they represent the state-of-the-art in this field.
> * **Broader Contribution:** The SCORE framework is not restricted to e-LORD and e-SAFFRON. We utilized them primarily as canonical examples to demonstrate a much broader, fundamental theoretical advance. Our work introduces a universal ''plug-in'' framework that improves the foundational mathematical machinery of the field. Almost all online multiple testing algorithms follow the same overarching paradigm:
>     1. Construct an  FDP estimator $\widehat{\text{FDP}}(t)$ which satisfies $\text{FDR}(t) \le \mathbb{E}[\widehat{\text{FDP}}(t)]$.
>     2. Construct and adjust the threshold $\alpha_t$ to bound this estimator below the target level $\alpha$.
>
> 	Fundamentally, existing e-value-based procedures rely on the standard Markov's inequality to prove validity in Step 1. The SCORE framework advances the state-of-the-art by replacing this ubiquitous, loose inequality with our overshoot refund mechanism. As formalized in Theorem 3.2, SCORE strictly tightens the FDP estimator, establishing the inequality:
> 	$\text{FDR}(t) \le \mathbb{E}[\widehat{\text{FDP}}_{\text{SCORE}}(t)] \le \mathbb{E}[\widehat{\text{FDP}}\_{\text{baseline}}(t)]$
> By safely reducing the estimated error, it frees up more of the $\alpha$ budget. As formalized in Theorem 3.2, any current or future e-value-based algorithm following this standard architecture can be uniformly improved by plugging in this tighter estimator.
>
> **3. Regarding the method's behavior under strong or challenging evidence, and regimes with the most practical benefits:**
>
> Your intuition is absolutely correct. If every alternative is accompanied by very strong evidence (meaning the conventional method already achieves near-perfect power), our advantage is limited. However, in practical applications, it is rare for any method to consistently achieve near-perfect power.
>
> Regarding your question on challenging alternatives, in the extreme scenario where every e-value falls below the rejection threshold, our method safely reduces to the conventional procedure without any penalty. However, it is worth noting that this scenario implies the baseline method's power is exactly zero, which rarely occurs in practice. Moreover, we emphasize that SCORE provides a uniform improvement over the baseline. As theoretically established in Propositions 4.2, 4.3, and 4.4, if even a single past hypothesis is rejected, our adjusted wealth mechanism strictly lowers the rejection thresholds for all subsequent tests.
>
> Consequently, the specific scenario you highlighted regarding evidence close to the rejection threshold is precisely where our framework proves most valuable. If the data stream contains strong signals early on, the recycled wealth strictly lowers the threshold. This gives those borderline alternatives a much higher chance to be detected. Therefore, the regime where​ the proposed method provides the most practical benefits is precisely the one in which​ a large number of alternative hypotheses are borderline.
>
> **References:**
>
> [1] Xu, Ziyu, and Aaditya Ramdas. "Online multiple testing with e-values." *International Conference on Artificial Intelligence and Statistics*, pp. 3997-4005. PMLR, 2024.
>
> [2] Zhang, Yifan, Zijian Wei, Haojie Ren, and Changliang Zou. "e-GAI: e-value-based Generalized $\alpha$-Investing for Online False Discovery Rate Control." *Forty-second International Conference on Machine Learning*. 2025.

---

> > ### Author Rebuttal · Reviewer_31Fr · 2026-04-03
> >
> > Thank you for your response. I will keep my positive score.

---

> > > ### Author Response · Authors · 2026-04-07
> > >
> > > Thank you very much for your positive feedback and for confirming that your concerns have been fully addressed. We appreciate your valuable time and your professional support for our submission.

---

### Decision · Program_Chairs · 2026-04-30

**Decision:**

Accept (regular)

**Comment:**

This paper addresses the problem in online multiple hypothesis testing where e-value-based methods fail to fully utilize evidence exceeding the rejection threshold, leading to a loss of statistical power. To tackle this issue, the authors propose a unified framework that reuses the excess evidence beyond the threshold to improve efficiency. The proposed approach consistently enhances existing methods while preserving valid finite-sample FDR control, enabling more aggressive and powerful testing strategies. All reviewers recognize the novelty and significance of the proposed method. Overall, this work represents an important contribution to online multiple hypothesis testing by improving power while maintaining rigorous error control.